# Exploring cellular changes in ruptured human quadriceps tendons at single-cell resolution

Jolet Y. Mimpen[1,2] , Mathew J. Baldwin[1], Claudia Paul[1], Lorenzo Ramos-Mucci[1],
Alina Kurjan[1], Carla J. Cohen[1,3], Shreeya Sharma[1], Marie S. N. Chevalier Florquin[4],
Philippa A. Hulley[1], John McMaster[5], Andrew Titchener[5], Alexander Martin[5], Matthew L. Costa[1,5],
Stephen E. Gwilym[1,5], Adam P. Cribbs[1,6] and Sarah J. B. Snelling[1]

[1]*The Botnar Institute of Musculoskeletal Sciences, Nuffield Department of Orthopaedics Rheumatology and Musculoskeletal Sciences, University of Oxford, Oxford, UK*
[2]*Kennedy Institute of Rheumatology, Nuffield Department of Orthopaedics Rheumatology and Musculoskeletal Sciences, University of Oxford, Oxford, UK*
[3]*Centre for Computational Biology, MRC Weatherall Institute of Molecular Medicine, University of Oxford, Oxford, UK*
[4]*Leiden University Medical Center, Leiden University, Leiden, the Netherlands*
[5]*Oxford University Hospital NHS Foundation Trust, Oxford, UK*
[6]*Oxford Centre for Translational Myeloma Research University of Oxford, Oxford, UK*

Handling Editors: Karyn Hamilton & Martino Franchi

The peer review history is available in the Supporting Information section of this article (https://doi.org/10.1113/JP287812#support-information-section).

**Abstract figure legend** This study explores the cellular landscape of healthy and ruptured quadriceps tendons using single nucleus RNA sequencing. While a range of stromal and immune cell types and subsets were identified, the data indicate that fibroblasts and endothelial cells are the main drivers of the early injury response within ruptured quadriceps tendon. These cell types make up the majority of the cells within both healthy and ruptured quadriceps tendon, show the highest number of cell–cell interactions, and shift to an activated phenotype following rupture, pointing towards a fibrotic and angiogenic response. Therefore, these activated stromal cell types are potential targets for interventions to enhance tendon healing. Overall, this study highlights that the development of more effective therapeutic options for tendon injury requires better understanding of the cellular, extracellular, and mechanical landscape of tendon tissue, particularly across the early to late phases following injury. Created with BioRender.com.

J. Y. Mimpen and M. J. Baldwin contributed equally to this work.

This article was first published as a preprint. Mimpen JY, Baldwin MJ, Paul C, Ramos-Mucci L, Kurjan A, Cohen CJ, Sharma S, Chevalier Florquin MSN, Hulley PA, McMaster J, Titchener A, Martin A, Costa ML, Gwilym SE, Cribbs AP, Snelling SJB. 2024. Exploring cellular changes in ruptured human quadriceps tendons at single-cell resolution. bioRxiv. https://doi.org/10.1101/2024.09.06.611599

The Journal of Physiology

**Abstract** Tendon ruptures in humans have often been studied during the chronic phase of injury, particularly in the context of rotator cuff disease. However, the early response to acute tendon ruptures remains less investigated. Quadriceps tendons, which require prompt surgical treatment, offer a model to investigate this early response. Therefore, this study aimed to explore the early cellular changes in ruptured compared to healthy human quadriceps tendons. Quadriceps tendon samples were collected from patients undergoing fracture repair (healthy) or tendon repair surgery (collected 7–8 days post-injury). Nuclei were isolated for single-nucleus RNA sequencing, and comprehensive transcriptomic analysis was conducted. The transcriptomes of 12,808 nuclei (7268 from healthy and 5540 from ruptured quadriceps tendons) were profiled, revealing 12 major cell types and several cell subtypes and states. Rupture samples showed increased expression of genes related to extracellular matrix organisation and cell cycle signalling, and a decrease in expression of genes in lipid metabolism pathways. These changes were predominantly driven by gene expression changes in the fibroblast, vascular endothelial cell (VEC), mural cell, and macrophage populations: fibroblasts shift to an activated phenotype upon rupture and there is an increase in the proportion of capillary and dividing VECs. A diverse immune environment was observed, with a shift from homeostatic to activated macrophages following rupture. Cell–cell interactions increased in number and diversity in rupture, and primarily involved fibroblast and VEC populations. Collectively, this transcriptomic analysis suggests that fibroblasts and endothelial cells are key orchestrators of the early injury response within ruptured quadriceps tendon.

(Received 4 October 2024; accepted after revision 21 February 2025; first published online 14 April 2025)
**Corresponding author** J. Y. Mimpen: Nuffield Department of Orthopaedics, Rheumatology, and Musculoskeletal Sciences, Botnar Institute of Musculoskeletal Sciences, University of Oxford; Old Road, Headington, Oxford, OX3 7LD, UK. Email: jolet.mimpen@ndorms.ox.ac.uk

### Key points

- Tendon ruptures in humans have regularly been studied during the chronic phase of injury, but less is known about the early injury response after acute tendon ruptures.
- This study explored the early cellular changes in ruptured compared to healthy human quadriceps tendons at single-cell resolution.
- Fibroblasts and endothelial cells seem to be the key orchestrators of the early injury response within ruptured quadriceps tendon. Therefore, these cell types are obvious targets for interventions to enhance tendon healing.
- Overall, this study highlights that the development of more effective therapeutic options for tendon injury requires better understanding of the cellular, extracellular, and mechanical landscape of tendon tissue.

## Introduction

Tendons are complex hierarchical structures of extracellular matrix (ECM) that act to efficiently transfer tensile forces from the muscles to the skeleton. Tendon injuries resulting in partial or complete tendon rupture are common, particularly in ageing individuals. Tendon rupture frequently results in significant pain and disability,

**Jolet Mimpen** is a Research Fellow at NDORMS, University of Oxford, Oxford, UK. Her work focuses on tendon biology and osteoarthritis pathophysiology. **Mathew Baldwin** is a NIHR Academic Clinical Lecturer at NDORMS, University of Oxford, Oxford, UK. His work is focussed on translational orthopaedics. Dr Mimpen, Dr Baldwin, Prof. Sarah Snelling, and their colleagues collaborate on several projects, including multiple atlasing projects of musculoskeletal tissues for the Human Cell Atlas. Throughout this process, they have discovered that multidisciplinary teamwork and perseverance are the key to success.

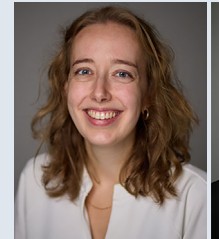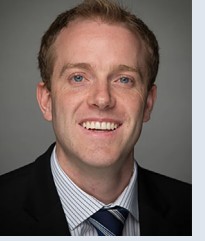

leading to a reduced quality of life. Furthermore, the incidence and socioeconomic costs of these injuries are increasing (Millar et al., 2021).

Acute tendon ruptures are hypothesised to occur on the background of pre-existing tendinopathy and usually require surgical intervention or prolonged immobilisation (Maffulli et al., 2012; Trobisch et al., 2010). Tendons often heal poorly, resulting in weaker tissues that prevent return to pre-injury activities (Ciriello et al., 2012). This is due to the formation of granulation (scar) tissue during proliferative stages of tendon healing that fails to recapitulate the mechanical, structural and chemical properties of healthy tendon ECM. These weak tendons are more susceptible to re-rupture or can progress to chronic tendon disease and disability. To date, a limited understanding of the cell types that contribute to both tendon homeostasis and early injury response has hampered our ability to biologically augment tendon repair and improve these outcomes. Importantly, transcriptomic mapping of tendons at single-cell resolution has recently allowed the cellular and molecular landscapes of healthy human hamstring tendons to be deciphered (Akbar et al., 2021; Karlsen et al., 2023; Kendal et al., 2020; Mimpen et al., 2024).

While the overall cellular content of healthy human tendon tissue is low, several cell types can be found such as fibroblasts, endothelial cells, and certain types of immune cells (Akbar et al., 2021; Fu et al., 2023; Karlsen et al., 2023; Kendal et al., 2020; Mimpen et al., 2024). Fibroblasts, in tendon sometimes referred to as tenocytes, are responsible for ECM production and maintenance in tendon development, homeostasis and repair (McNeilly et al., 1996; Mimpen et al., 2024). In adult murine models, the origin of the fibroblasts that contribute to early tendon repair has been attributed to epitenon or perivascular derived cells, while immune cells, particularly macrophages, have been implicated as major orchestrators of both the acute and the chronic response to tendon injury (Crosio & Huang, 2022; Nichols et al., 2023). Fibroblast activation has been studied in human chronic tendinopathy, particularly in rotator cuff tendons of the shoulder (Crowe et al., 2021; Dakin et al., 2017; Dean et al., 2012; Millar et al., 2021), in which tears are often atraumatic and might be surgically repaired months or years after a non-injury event (Abdul-Wahab et al., 2016; Dang & Davies, 2018; Keener et al., 2015). However, the fibroblast cell types and states important in the early human response to acute tendon rupture have not been characterised well. Similarly, the role of the immune, endothelial, and mural cell compartments has been investigated in the context of chronic tendinopathy in humans and rarely during the early injury response of an acute tendon injury (Akbar et al., 2021; Fu et al., 2023; Maffulli et al., 2012). The paucity of knowledge on the cell types acting early in the tendon repair response limits progress in improving patient outcomes following tendon rupture.

This study aimed to identify the cellular components of the early tendon response to rupture. To enable this, we undertook single nucleus RNA sequencing (snRNA-seq) on ruptured and healthy human quadriceps tendons. Quadriceps tendons were used for this study since they are one of the most common tendons to undergo repair for acute rupture. Samples of quadriceps tendon ruptures, retrieved 7–8 days post-injury, allowed exploration of the cellular and molecular characteristics of the fibroproliferative phases of early human tendon injury.

## Methods

### Ethics

Ethical approval was granted for the Oxford Musculoskeletal Biobank (19/SC/0134) by the local research ethics committee (Oxford Research Ethics Committee B) for all work on healthy and ruptured human quadriceps tendon, and informed written consent was obtained from all patients according to the *Declaration of Helsinki*.

### Tissue acquisition and processing

Healthy quadriceps tendon tissue was collected from patients undergoing suprapatellar intramedullary nail surgery for tibial shaft fracture within 2 days after injury or above the knee amputation for malignant disease. Ruptured quadriceps tendon tissue was collected from patients undergoing tendon repair surgery for complete quadriceps tendon rupture 7–8 days post-injury. All tissues used in this work came from the midbody of the tendon. Patients were not included in the study if they had an active infection at time of surgery. The age, sex, and self-reported ethnicity of donors, alongside the affected side of the injury, were collected and are reported along the de-identified study ID of each patient (Table 1). Tendon tissue was processed as previously described (Mimpen et al., 2022). Briefly, tissue was collected in cold Dulbecco's modified Eagle's medium (DMEM)–F12 medium (Thermo Fisher Scientific) supplemented with 10% fetal bovine serum (Labtech International, Heathfield, UK) and 1% penicillin–streptomycin (Thermo Fisher Scientific, UK). The collected sample was processed within 2 h of tissue collection. Tendon was washed in phosphate-buffered saline (PBS), fat and muscle were dissected off, and the tendon was cut into 1 cm pieces and photographed to retain topographical reference. The pieces of tendon tissue were snap-frozen in cryotubes using liquid nitrogen and stored at −80°C until use.

**Table 1. Summary of patient demographics, including de-identified patient ID, sex, age (in years), self-reported ethnicity, affected side, diagnosis, and surgery type**

| Group | Patient ID | Sex | Age (yrs) | Self-reported ethnicity | Affected side | Diagnosis | Surgery type |
|---|---|---|---|---|---|---|---|
| Healthy | MSK 0792 | Male | 29 | White – British | Right | Tibial shaft fracture | Suprapatellar IM nail surgery |
| | MSK 1248 | Male | 44 | White – other | Right | Tibial shaft fracture | Suprapatellar IM nail surgery |
| | MSK 1250* | Male | 45 | White – other | Left | Malignant disease | Above the knee amputation |
| | MSK 1266 | Male | 25 | White – British | Unknown | Tibial shaft fracture | Suprapatellar IM nail surgery |
| Rupture | MSK 0778 | Male | 67 | White – other | Left | Quadriceps rupture | Tendon repair |
| | MSK 0779 | Male | 75 | White – British | Right | Quadriceps rupture | Tendon repair |
| | MSK 0793 | Male | 69 | White – British | Right | Quadriceps rupture | Tendon repair |

Note: *MSK 1250 was not included in the final analysis due to high skeletal muscle contamination (Supporting information, Supplementary Fig. S1).

## Single nucleus RNA sequencing

**Isolation of nuclei.** Nuclei were isolated from the snap-frozen tissue using our previously published protocol (Mimpen et al., 2021). In short, quadriceps tendon (midbody) was cut into thin sections using a scalpel, forceps and a Petri dish that were all pre-cooled on dry ice and stored in a 50 ml Falcon tube at −80°C until use. On the day of cell lysis, the tubes containing the cut quadriceps tendons were thawed, and 4 ml of cold $1\times$ CST buffer [containing NaCl, Tris–HCl pH 7.5, $CaCl_2$, and $MgCl_2$, with CHAPS hydrate (Sigma-Aldrich, Merck Life Sciences, Darmstadt, Germany), BSA (Sigma-Aldrich), RNase inhibitors (RNaseIn Plus (Promega, Madison, WI, USA) and SUPERase In (Thermo Fisher Scientific), and protease inhibitor (cOmplete tablet, Roche Diagnostics GmbH, Mannheim, Germany); the full recipe can be found in the published protocol] was added. After 10 min of incubation on a rotor at 4°C, the tissue/buffer mixture was poured through a 40 μm strainer and the tube used for tissue lysis was washed twice with 2 ml PBS with 1% BSA. The nuclei solution was then transferred to a 15 ml Falcon tube, and the previous 50 ml tube was washed once with 4 ml PBS with 1% BSA and transferred to the same 15 ml Falcon tube. The nuclei solution was centrifuged at 500 $g$ at 4°C for 5 min. After pouring off the supernatant, the tubes were briefly spun down, the nuclei were resuspended in the supernatant, and the remaining volume was determined. Concentration of nuclei was determined counting 4′,6-diamidino-2-phenylindole (DAPI)-stained nuclei in a Neubauer Improved haemocytometer (NanoEnTek, Seoul, Korea).

**Library preparation and sequencing.** Nuclei suspensions were diluted (PBS with 1% BSA) to 240–1000 nuclei/μl and loaded on the Chromium Next GEM Chip G (10x Genomics, Pleasanton, CA, USA) with a targeted nuclei recovery of 600–10,000 nuclei per sample. Samples were then loaded onto the Chromium Controller (10x Genomics) and libraries were prepared using the Chromium Next GEM Single Cell 3′ Reagent Kits v3.1 (10x Genomics) following the manufacturer's instructions and indexed with the Single Index Kit T Set A (10x Genomics). Quality control of cDNA and final libraries was analysed using High Sensitivity ScreenTape assays on a 4150 TapeStation System (Agilent Technologies, Santa Clara, CA, USA). Final libraries were pooled and sequenced on a NovaSeq 6000 (Illumina, San Diego, CA, USA) by Azenta Life Sciences (GENEWIZ from Azenta Life Sciences, South Plainfield, NJ, USA) at a minimum depth of ∼20,000 read pairs per expected nuclei.

## Single nucleus RNA sequencing data analysis

**Quality control, integration, clustering and annotation.** Raw next generation sequencing (NGS) data were quality controlled and mapped using the *scflow quantnuclei* pipeline (https://github.com/cribbslab/scflow; python v3.8.15). Specifically, quality control of Fastq files was performed using fastqc (v0.12.1). Fastq files were mapped to the human Ensembl GRCh38 transcriptome (release 106) using kallisto bustools kb count (v0.27.3) with kmer size = 31 (https://www.kallistobus.tools/kb_usage/kb_count/). Spliced and unspliced matrices were merged to create the

**Table 2. Quality control of snRNA-seq samples, including the minimum number of UMI counts (nCount), minimum number of unique genes (features) (nFeature), maximum percentage of mitochondrial reads and final number of nuclei after all filtering (including removal of doublets and droplets with high ambient RNA contamination)**

| Group | Patient ID | nCount | nFeature | mito_percent | Nuclei after filtering |
|---|---|---|---|---|---|
| Healthy | MSK 0792 | 500 | 500 | 5 | 1014 |
| | MSK 1248 | 500 | 500 | 5 | 4831 |
| | MSK 1266 | 1000 | 500 | 5 | 1423 |
| Rupture | MSK 0778 | 500 | 500 | 5 | 3679 |
| | MSK 0779-1* | 300 | 300 | 5 | 661 |
| | MSK 0779-2* | 300 | 300 | 5 | |
| | MSK 0793 | 500 | 500 | 5 | 1200 |

Note: MSK 1250 was not included in the final analysis due to high skeletal muscle contamination (Supporting information, Supplementary Fig. S1). *Due to low nuclei yield, a second piece of tissue from the same donor was prepared for a second run and sequenced separately. Nuclei from both runs were combined for donor MSK 0779.

count matrix for downstream analysis. The snRNA-seq analysis was performed in R (v4.3.1) and RStudio Server (v2023.03.1 build 446) using Seurat (v4.3.0.1) (Hao et al., 2021). Filtering thresholds for number of cells (nCount), number of features (nFeature) and mitochondrial ratio (mitoRatio) were set manually for each sample to remove poor-quality cells (Table 2 and Supporting information, Supplementary Fig. S2). Ambient RNA was detected using the decontX function from the celda package (v1.14.0) and SoupX package (v1.6.2). SoupX was run using the estimateNonExpressingCells() and the calculateContaminationFraction() function using a list of either macrophage markers (*CD163*, *CD163L1*, *CD14*, *MRC1*, *MSR1*) or fibroblast markers (*COL1A1*, *COL1A2*, *COL3A1*, *DCN*) to estimate the non-expressing cells (Young & Behjati, 2020). Doublets were detected with scDblFinder (v1.12.0) using the default settings (Germain et al., 2022). Doublets and cells with high decontX_contamination scores ($>0.3$) were removed. All datasets were merged, SoupX-adjusted counts matrix (SoupXcounts) of the merged data were log-normalised, 5000 variable features were selected, the data were scaled, and 50 principal components (PCs) were calculated. Integration of samples was performed using Harmony (v0.1) implemented within the Seurat workflow using each sample ID as the confounder ('orig.ident'). Integrated data were clustered using the FindNeighbors(), FindClusters(), and RunUMAP() functions using 40 dimensions and 0.2 resolution. DotPlot() and DoHeatmap() functions from the Seurat package, as well as the EnhancedVolcano, pheatmap, and ggplot packages were used to visualise the data. Broad cell-type annotations were assigned according to expression of canonical marker genes described in the literature.

**Single nucleus RNA-seq data analysis – differential abundance analysis using MiloR.** Differential abundance testing of cell types between healthy and rupture samples was performed using MiloR (version 1.10.0) (Dann et al., 2021). Analysis was performed using either broad cell type annotations or a subset of finely annotated cells. The *k*-nearest neighbour (KNN) graph of cells was constructed on Harmony embeddings using parameters $k = 30$ and $d = 30$, and neighbourhoods were defined using a random sampling rate of 0.1. A negative binomial linear regression model was used to assess the effect of disease status on the number of cells in each neighbourhood from each donor, and correction for multiple testing was performed using weighted Benjamini–Hochberg correction. Neighbourhoods were classified according to the cell type that contributed the majority of cells that comprised the neighbourhood ($>70\%$, otherwise classed as 'mixed').

**Pseudobulk differential gene expression analysis and pathway analysis.** Pseudobulk analysis was used to determine differential gene expression (DGE) between healthy and ruptured quadriceps tendons, either across the whole sample or per cell type using the AggregateExpression() function, with genes with less than 10 reads filtered out. DESeq2 was used for the DGE analysis with the design formula specified as '$\sim$ tendon_disease' to split the data into a healthy and a rupture group. The apeglm method was used for log fold change (LFC) shrinkage (Love et al., 2014). Results were visualised using ggplot2 and EnhancedVolcano packages. After excluding any genes for which a *P*-value could not be calculated (NA), genes with an absolute log2 fold change value of at least 0.58 (at least 1.5-fold increase or decrease) after lfc shrinkage were used for pathway analysis. Hallmark and Gene Ontology biological process (GO:BP) pathway analysis was done using fgsea. Pathway analysis results were visualised using enrichplot.

**Table 3. The selected number of variable features, dimensions and resolution, as well as the total nuclei number for each group of cells that was subclustered**

|  | Variable features | Dimensions | Resolution | Total nuclei number |
|---|---|---|---|---|
| Fibroblasts | 2000 | 20 | 0.2 | 5791 |
| Endothelial cells | 1000 | 30 | 0.5 | 2844 |
| Immune cells | 1500 | 20 | 1.1 | 3178 |

**Fine annotation and pathway analysis.** Fine annotation of cell types and states was performed by subclustering of broad cell types. Fibroblasts, including fibroblasts and 'dividing fibroblasts/mural cells', endothelial cells, including vascular endothelial cells, lymphatic endothelial cells, and mural cells, or all immune cells, including dendritic cells, dividing macrophages, granulocytes, macrophages, osteoclasts and T cells, were subsetted. Each subset was log-normalised, variable features were selected, the data were scaled, and 50 PCs were calculated. Integration of samples was performed using Harmony (v0.1) implemented within the Seurat workflow to remove the effect of the different samples ('orig.ident'). The chosen number of variable features, dimensions, resolution, and total number of cells for each group can be found in Table 3.

For GO:BP pathway analysis, the top 100 positive differentially expressed genes with a log2 fold change of at least 0.5 for each cluster were selected using the FindMarkers() function. GO:BP analysis was done using the gost() function from gprofiler2 (Kolberg et al., 2023). Annotation was performed using differentially expressed genes and canonical genes from the literature. Finally, the new annotations of the clusters found in fibroblasts, endothelial cells and immune cells were projected onto the original UMAP.

**Ligand–receptor interactions and activity interference analysis.** SCpubr was used for pathway activity inference analysis (using the PROGENy database) (Blanco-Carmona, 2022; Schubert et al., 2018), transcription factor activity interference analysis (using the dorothea database) (Badia-I-Mompel et al., 2022), and correlation matrix heatmap using the recommended settings as per the vignette for SCpubr v1.1.2. Liana was used for ligand–receptor analysis, using the CellPhoneDB database and selecting interactions with a *P*-value of at least 0.05 (Dimitrov et al., 2022). The CrossTalkeR package was used to visualise ligand–receptor interaction results using the plot_cci() function (Nagai et al., 2021).

**SCENIC analysis.** Single-cell regulatory network inference and clustering (SCENIC) analysis was carried out on the command line and in a Jupyter Notebook in an anaconda environment with Python v3.7.12 and pyscenic v0.11.2, among other packages. Briefly, harmony-integrated Seurat data were converted into the h5ad format using Sceasy (Cakir et al., 2020). The raw, un-normalised counts matrix as well as the accompanying anndata observations and variables were made into a new object and saved as a.loom file, which was used as input for the pySCENIC workflow steps (Aibar et al., 2017; van de Sande et al., 2020). Gene regulatory network (GRN) inference was carried out using the GRNBoost2 algorithm by running the 'pyscenic grn' command using default settings. A predefined list of human transcription factors used for this network inference step was retrieved from the Aertslab github repository (https://github.com/aertslab/SCENICprotocol/tree/master/example, 'allTFs_hg38.txt', last updated 3 years ago). The resulting list of transcription factor–gene interactions was then used to infer transcription factor–gene co-expression modules, identify enriched motifs, and predict regulons by running the 'pyscenic ctx' command with default settings. Input genome rankings.feather and motif annotation.tbl files were retrieved from the Aertslab cistarget resources webpage (v9 files; https://resources.aertslab.org/cistarget/). In total, 317 regulons were identified. Next, the activity of predicted regulons in individual cells was quantified by running the 'pyscenic aucell' command using default settings. For each regulon, cellular regulon activity was then binarised as 'on' or 'off' using a Gaussian mixture model. Additionally, we assessed regulon-cell type cluster specificity using the regulon specificity score (RSS).

## Results

### Single nucleus RNA sequencing defined cellular and molecular profiles of acute tendon injury

To define the cellular and molecular profiles of acute tendon injury we collected snap-frozen tissue samples from four healthy and three ruptured human quadriceps tendons. After extensive quality control, the transcriptomes of 12,808 nuclei were profiled, including 7268 nuclei from three healthy donors (male, 25–44 years old) and 5540 nuclei from three ruptured quadriceps tendon donors (male, 67–75 years old). Integration

was performed using Harmony across all samples. Following unsupervised clustering, we identified 14 clusters, which were annotated based on the expression of canonical markers (Fig. 1 and Supporting information, Supplementary Fig. S3). These included fibroblasts (*COL1A2, COL3A1, DCN*), macrophages (*CD163, MRC1, MSR1*), vascular endothelial cells (VECs) (*PECAM1, PTPRB, VWF*), mural cells (*NOTCH3, PDGFRB, MYO1B*), adipocytes (*PLIN1, PLIN4, ADIPOQ*), T cells (*CD247, SKAP1, THEMIS*), nervous system cells (*NRXN1, CADM2, IL1RAPL2*), lymphatic endothelial cells (*MMRN1, PROX1, PKHD1L1*), dividing fibroblasts/mural cells and dividing macrophages (*ASPM, DIAPH3, TOP2A*; more in Supporting information, Supplementary Fig. S4), dendritic cells (DCs) (*CLEC4C, CUX2, BCL11A*), osteoblasts (*IBSP, SP7, RUNX2*), granulocytes (*KIT, CPA3, IL18R1*) and osteoclasts (*MMP9, ACP5, SIGLEC15*).

Differences were observed in the mean cell type frequency between healthy and ruptured quadriceps tendons (Fig. 1*C*), and the trends in cell type abundance were similar across patients (Supporting information, Supplementary Fig. S5). To test for differences in cellular composition between healthy and ruptured tendons, MiloR was used to assign neighbourhoods and test these neighbourhoods for differential abundance. Several neighbourhoods had a different cellular composition in ruptured compared to healthy quadriceps tendons, including increases in the abundance of osteoblasts and dividing fibroblasts/mural cells and decreases in the abundance of lymphatic endothelial cells and adipocytes (Fig. 1*D* and Supporting information, Supplementary Fig. S6). Fibroblasts, VECs, and mural cells all had neighbourhoods that were increased or decreased in abundance in rupture, warranting further investigation of potential subsets of these cell types.

To understand the overall and cell type-specific differences in gene expression between ruptured and healthy quadriceps tendons, we aggregated the snRNA-seq counts by patient (overall differences) or by patient and cell type (cell type-specific differences) and performed DGE analysis. In total, this analysis revealed 405 differentially expressed genes ($P_{adj} < 0.05$, log2 fold change $\pm 1$) in ruptured compared to healthy quadriceps tendons, including increases in *COL1A1, COL1A2, COL3A1, COL6A3, FAP* and *SPP1* (Fig. 2*B* and *C*). Gene ontology biological process (GO:BP) analysis of these differences revealed a significant decrease in pathways relating to 'lipid acid metabolic pathways' and 'actomyosin structure organisation' in ruptured quadriceps tendons, while increases were found in pathways relating to 'collagen fibre organization', 'bone morphogenesis', 'biomineralisation', and 'cell cycle checkpoint signalling' (Fig. 2*D*). DGE analysis comparing ruptured to healthy quadriceps tendons within each individual cell type revealed that most differentially expressed genes could be found in fibroblasts (586), VECs (386), and mural cells (185). Few or no changes were found within the macrophages (43), T cells (6), and dividing macrophages (0), while other cell types could not be compared due to low cell numbers or complete absence of these cell types in at least one of the donors (Fig. 2*B*).

## Fibroblasts in ruptured quadriceps tendon switch to an ECM-producing phenotype

The highest number of differentially expressed genes between healthy and ruptured quadriceps tendons were found in the fibroblast population (Fig. 3*A*). Sixty-four out of the 586 (10.9%) differentially expressed genes are part of the matrisome – a collection of core (collagens, glycoproteins, proteoglycans) and associated ECM proteins (Fig. 3*B*). All 10 collagen genes that were differentially expressed were increased in rupture samples, which include *COL1A1, COL1A2* and *COL3A1*; other highly increased ECM genes include *POSTN, ADAM12, TGFBI, SPARC* and *TNC*. ECM genes that were higher in healthy fibroblasts include *TNXB, HPSE2, MMP3, FBLN1* and *LAMA2*. Significant differentially expressed genes ($P_{adj} < 0.05$ and LFC $\pm0.58$) within fibroblasts were used for hallmark pathway analysis, which showed a decrease in adipogenesis, bile acid metabolism and fatty acid metabolism, and an increase in epithelial–mesenchymal transition (EMT) pathways in ruptured compared to healthy quadriceps tendons (Fig. 3*C*).

Fibroblasts and dividing fibroblasts/mural cells were subsetted, re-clustered, and re-annotated, revealing completely distinct transcriptional phenotypes between healthy and ruptured tendons (Fig. 3*D, E* and Supporting information, Supplementary Fig. S7). Three different fibroblasts subsets were found in healthy quadriceps tendons: FBLN1$^{hi}$ fibroblasts (*FBLN1, NOX4, CILP, COMP*), ABCA10$^{hi}$ fibroblasts (*ABCA10, CNTN4, C6, ABCA8*), and NR4A1$^{hi}$ fibroblasts (*NR4A1, NR4A3, NAMPT, SEMA4A*). Ruptured quadriceps tendon fibroblasts were predominantly ADAM12$^{hi}$ fibroblasts (*ADAM12, COL3A1, TNC, POSTN*), as well as dividing cells (*DIAPH3, TOP2A, CENPE, ASPM*), which included both dividing fibroblasts and mural cells as defined in Fig. 1. Differential abundance analysis of cell neighbourhoods by MiloR confirmed the condition-specific presence of these fibroblasts within healthy or ruptured tendons (Fig. 3*G*).

Differences in the enriched GO:BP pathways based on the 100 most differentially expressed genes in each fibroblast cluster suggested different functional roles for these fibroblasts (Fig. 3*F*). Fibroblasts in healthy tendons were enriched for functions related to locomotion and

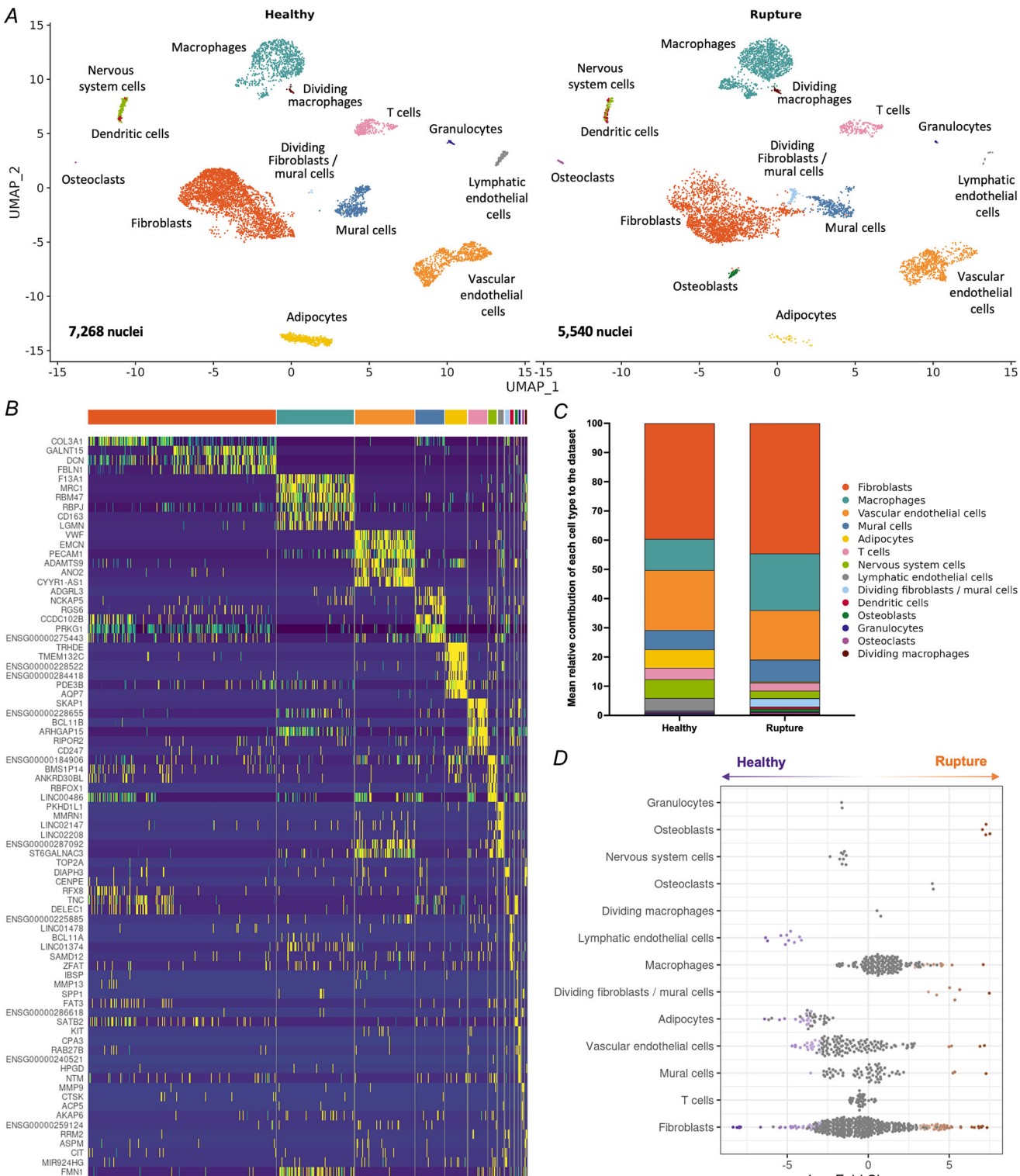

**Figure 1. Single nucleus RNA-seq analysis of healthy and ruptured quadriceps tendon**
*A*, uniform manifold approximation and projection (UMAP) embedding of 12,808 nuclei from six individuals split by disease state (healthy and ruptured) quadriceps tendons reveals 14 clusters. *B*, heatmap of the top six positive differentially expressed genes for each cluster using the 5000 variable and scaled features. If gene symbol names were not available, ensemble names were used. *C*, bar graph showing the mean cell type frequency for healthy and ruptured quadriceps tendon. *D*, differential abundance analysis of cell neighbourhoods showing changes in relative abundance, with neighbourhoods showing no statistically significant differences (grey), relative increases in rupture (orange), or relative increases in healthy (purple). [Colour figure can be viewed at wileyonlinelibrary.com]

cell motility (FBLN1^hi fibroblasts), system development and cell adhesion (ABCA10^hi fibroblasts), and cellular response to chemical stimulus or organic substance (NR4A1^hi fibroblasts). Fibroblasts from ruptured tendons were enriched for ECM and extracellular structure organisation (ADAM12^hi fibroblasts), and cell cycle process pathways (dividing cells). The change in ECM and extracellular structure organisation can also be illustrated by the expression of *COL1A2*, a major structural collagen of healthy tendon ECM, and scar-associated *COL3A1*,

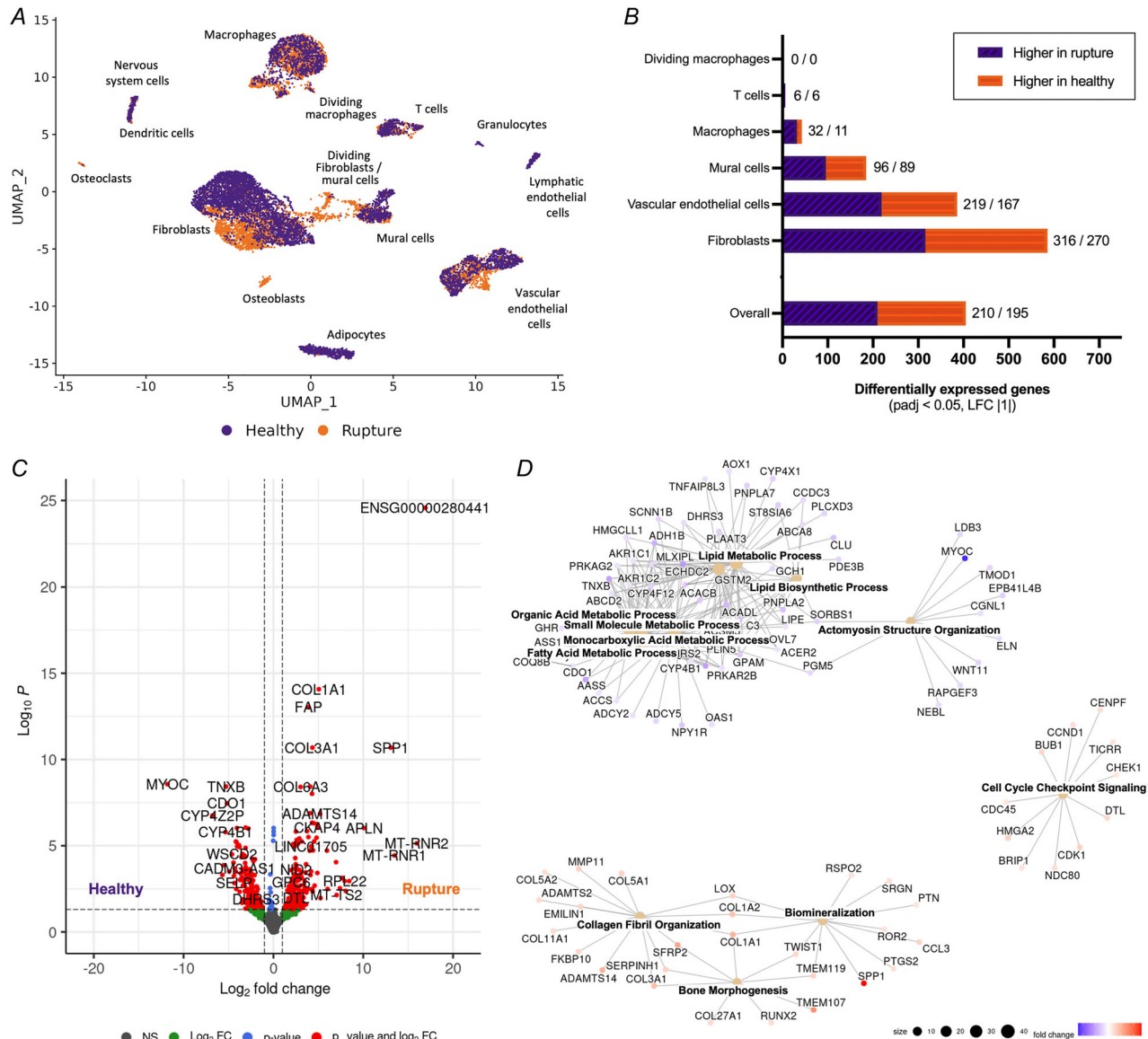

**Figure 2. Pseudobulk analysis of the overall datasets and identified clusters**
*A*, UMAP embedding of 12 808 nuclei from six individuals coloured by disease status (purple, healthy; orange, rupture). *B*, number of differentially expressed genes (higher in ruptured/higher in healthy) between healthy and ruptured tendon samples overall (counts aggregated by patients) or by cell type (counts aggregated by patient and cell type); adjusted *P*-value ($P_{adj}$) < 0.05 and log2 fold change (LFC) of at least ±1. *C*, volcano plot of overall differences in gene expression between healthy (left) and rupture (right) samples. Colours: grey: not-significant genes (NS); green: genes that reached the log2 fold change threshold, but not the *P*-value threshold; blue: genes that reached the *P*-value threshold but not the log2 fold change threshold; red: genes that were considered significant due to reaching both the *P*-value and log2 fold change threshold. *D*, gene-concept network plot of significantly changed gene ontology biological process (GO:BP) pathways using genes differentially expressed across all cell types between ruptured and healthy tendon. Genes included in the pathway analysis: $P_{adj}$ < 0.05, LFC ±0.58. Node size: number of genes in a category; colour: fold change of each displayed gene. [Colour figure can be viewed at wileyonlinelibrary.com]

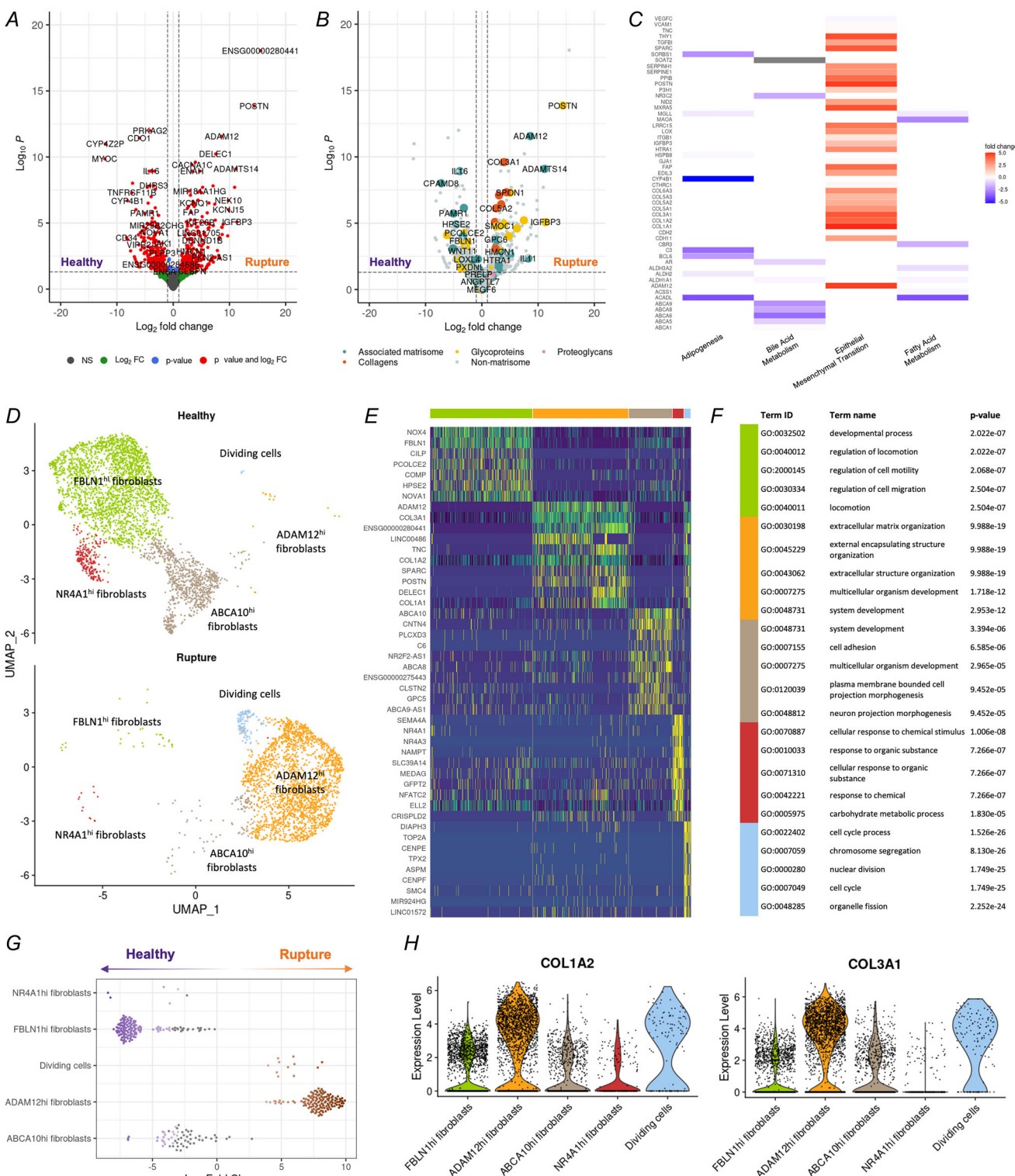

**Figure 3. The cellular state of fibroblasts shifts in ruptured quadriceps tendons**

*A* and *B*, volcano plots showing the differential gene expression results of healthy (left) *versus* ruptured (right) fibroblasts. *A*, colours refer to non-significant genes (NS; grey), genes that reached the log2 fold change threshold, but not the *P*-value threshold (green), genes that reached the *P*-value threshold but not the log2 fold change threshold (blue), or genes that were considered significant due to reaching both the *P*-value and log2 fold change threshold (red). *B*, colours refer to the type of matrisome component. *C*, hallmark pathways that were significantly changed in ruptured quadriceps tendon, based on genes with adjusted *P*-value ($P_{adj}$) (LFC) < 0.05 and log2 fold

change (LFC) ±0.58. *D*, UMAP embedding of sub-clustering of 5791 nuclei initially labelled as 'fibroblasts' or 'dividing fibroblasts/mural cells', labelled by their new cluster identity from healthy and rupture donors. *E*, heatmap with top 10 differentially expressed genes for each new fibroblasts cluster. *F*, top five enriched gene ontology biological process (GO:BP) pathways of each fibroblast subset based on the top 100 differentially expressed genes in each cluster. *G*, differential abundance analysis of cell neighbourhoods by MiloR showing relative increases in rupture (orange) or healthy (purple) cell neighbourhoods on a beeswarm plot. *H*, violin plots showing the expression of *COL1A2* and *COL3A1* across fibroblasts subsets. [Colour figure can be viewed at wileyonlinelibrary.com]

which were both strongly expressed in ADAM12$^{hi}$ fibroblasts of ruptured tendons (Fig. 3*H*).

## An increase in the abundance of capillary vascular endothelial cells in ruptured quadriceps tendons points to active angiogenesis in the early injury response

VECs and mural cells showed the second and third highest gene expression changes between ruptured and healthy quadriceps tendons (Fig. 2*B*), with GO:BP pathway analysis for VECs showing upregulation of several cell cycle pathways (Supporting information, Supplementary Fig. S8). Sub-clustering cells of the endothelium (VECs, lymphatic ECs, and mural cells) revealed four subsets of VECs and two subsets of mural cells (Fig. 4). VECs (*PECAM1*, *VWF*, *FLT1*) could be separated into venular (*ACKR1*, *LIFR*, *ICAM*), arteriolar (*NEBL*, CXCL12, *PDXL*), capillary (*COL4A1*, *COL4A2*, *DYSF*) and dividing VECs. Mural cells (*NOTCH3*, *PTPRB*) separated into pericytes (*ABCC9*, *EGFLAM*, *STEAP4*) and vascular smooth muscle cells (vSMCs) (*MYH11*, *ITGA8*, *NTRK3*). In ruptured quadriceps tendons, a relative increase in the proportion of capillary VECs, dividing VECs, and pericytes was found (Fig. 4*C*). MiloR relative abundance analysis confirmed that there was a significant increase in capillary and dividing VECs in rupture samples compared to healthy, pointing to an active angiogenesis response in the rupture samples (Fig. 4*D* and *E*). GO:BP pathway analysis of the top 100 differentially expressed genes of each cluster showed differences between the pericytes and vSMC clusters: while the pericytes were characterised by pathways previously seen in healthy fibroblasts, such as 'cell adhesion and migration' as well as 'structure morphogenesis' and 'collagen fibre organisation', the vSMCs were dominated by muscle pathways, including 'muscle system process', 'muscle contraction', 'muscle structure development', and 'muscle cell differentiation'. Differences between the capillary VECs compared to the venular and arteriolar VEC clusters were mainly driven by the activation of 'blood vessel development' and 'vasculature development' in capillary VECs (Fig. 4*F*). Inspection of the expression of pro-angiogenic factors (e.g. vascular endothelial growth factors (VEGFs), fibroblast growth factors (FGFs), neuro-philins (NRPs), transforming growth factor $\beta$ (TGF-$\beta$) proteins, platelet-derived growth factors (PDGFs),

and more) by all identified cell clusters revealed only small differences between healthy and rupture samples (Supporting information, Supplementary Fig. S9). *NRP1* (neurophilin 1, which is co-receptor for VEGF and semaphorin family members) was found to be highly expressed in VEC subsets, macrophages subsets as well as in some fibroblast subsets, while *NRP2* was mostly found in lymphatic ECs and osteoclasts. Expression of *NRP1* was slightly elevated in VEC and macrophage clusters of ruptured quadriceps tendon samples compared to healthy tendons. The VEC clusters were found to express *EPAS1*, *TEK*, *VEGFC*, *PDGFB*, *PDGFD*, *CXCL12*, *NOTCH1* and *EFNB2*. *HIF1A* was expressed in several fibroblast subsets, macrophages, DCs, osteoclasts, and vSMCs clusters in rupture samples.

## Quadriceps tendons have a diverse immune environment in health and following rupture

All immune cell clusters were re-clustered to enable fine annotation of the immune environment of quadriceps tendons. Sub-clustering of these cell types revealed different macrophage, DC, and lymphocyte subsets, as well as the previously identified granulocyte and osteoclast clusters (Fig. 5*A–C*).

Three macrophage subsets were identified in quadriceps tendons: MERTK$^{hi}$ LYVE1$^{hi}$, MERTK$^{hi}$ LYVE1$^{lo}$ and MERTK$^{lo}$ PTPRG$^{hi}$. Macrophages showed a shift from a high proportion of MERTK$^{hi}$ LYVE1$^{hi}$ in health to a high proportion of MERTK$^{hi}$ LYVE1$^{lo}$ in ruptured tendon samples (Fig. 5*D*); these MERTK$^{hi}$ LYVE1$^{lo}$ macrophages expressed high levels of *CSTB*, *TPTRG1*, *HMOX1*, and chemokines including *CXCL2*, *CXCL3* and *CXCL8*. DCs could be separated into CLEC10A$^{hi}$ DCs (*CLEC10A*, *CD1C*, *IL1R2*), CLEC9A$^{hi}$ DCs (*CLEC9A*, *IDO1*, *CLNK*), plasmacytoid DCs (pDCs) (*CLEC4C*, *IRF8*, *PLAC8*) and VCAN$^{hi}$ DCs/monocytes (*VCAN*, *FCN*, *LYZ*). Lymphocyte subsets included T cells (*IL7R*, *THEMIS*, *ANK3*), natural killer (NK) cells (*NCAM1*, *GNLY*, *KLRD1*) and B cells (*MS4A1*, *BLK*, *IGHM*). Finally, a group of dividing immune cells was found (*ASPM*, *DIAPH3*, *TOP2A*). Both pDCs and osteoclasts were exclusively found in ruptured quadriceps tendons.

A correlation matrix of the variable features showed similarities in the transcriptional profiles of T and NK cells, the four types of DCs, MERTK$^{hi}$ LYVE1$^{lo}$

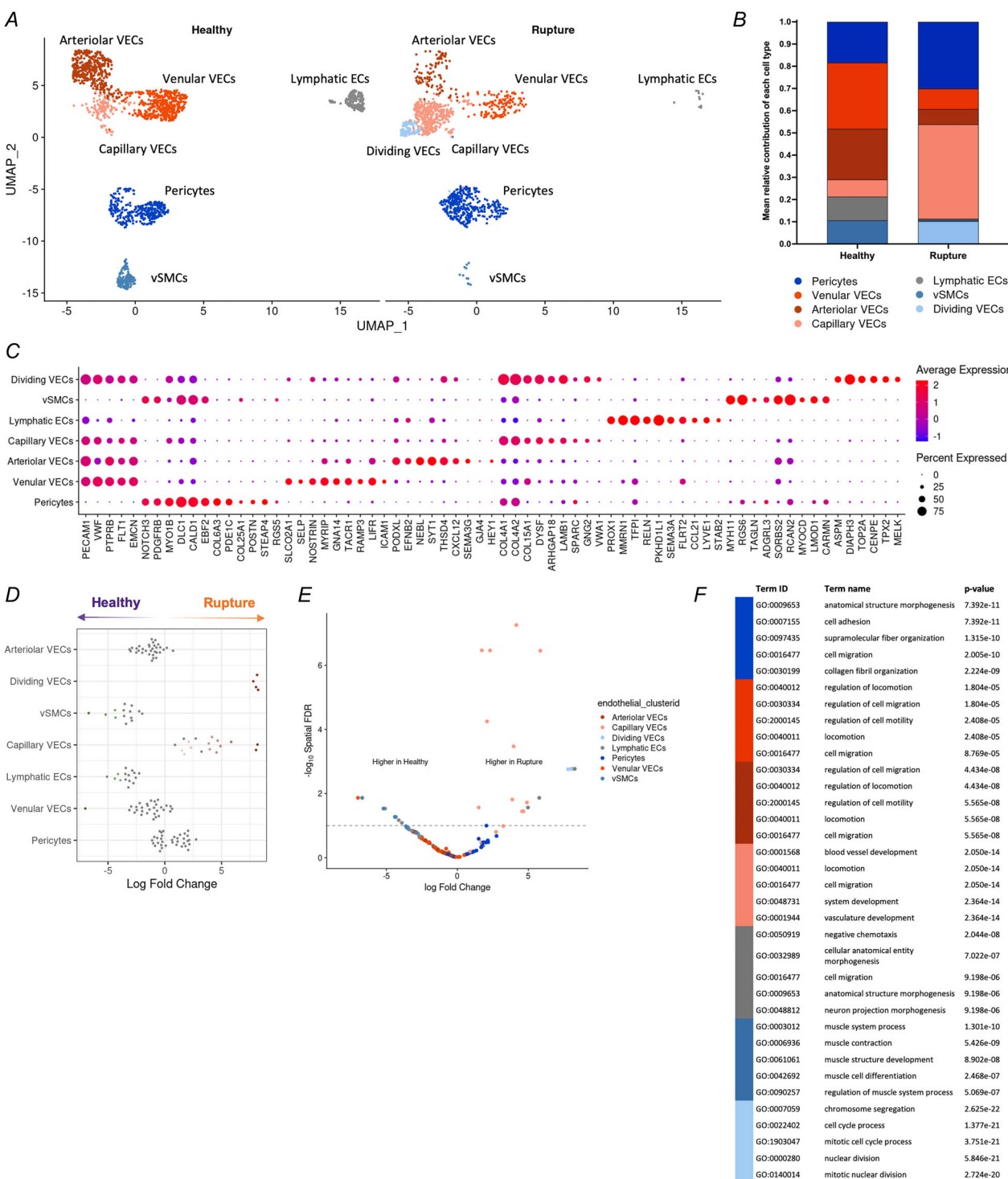

**Figure 4. Shift in proportion of endothelial subsets points to an active angiogenesis response**
*A*, UMAP embedding of sub-clustering of 2844 nuclei initially annotated as 'vascular endothelial cells' or 'lymphatic endothelial cells' or 'mural cells', labelled by endothelial fine annotation and split by tendon disease status. *B*, DotPlot with canonical and differentially expressed genes of different endothelial subsets. *C*, mean relative proportion of endothelial cell types in healthy and ruptured quadriceps tendons. *D* and *E*, differential abundance analysis of cell neighbourhoods showing relative increases in rupture (orange) or healthy (purple) cell neighbourhoods on a beeswarm plot (*D*) and plotted on a volcano plot coloured by endothelial subset (*E*). *F*, top five enriched gene ontology biological process (GO:BP) pathways of each endothelial subset based on the top 100 differentially expressed genes in each cluster. [Colour figure can be viewed at wileyonlinelibrary.com]

macrophages and dividing immune cells, and MERTK$^{hi}$ LYVE1$^{hi}$ macrophages and MERTK$^{lo}$ PTPRG$^{hi}$ macrophages (Fig. 5*E*). Changes in proportion of the different immune cell clusters were observed (Fig. 4*D*). Finally, while relative abundance analysis with MiloR showed an increase in MERTK$^{hi}$ LYVE1$^{lo}$ macrophages in ruptures, none of these changes were statistically significant (data not shown).

## Cell–cell interactions and intracellular activity is increased in the early stages following quadriceps tendon rupture

The finely annotated cell types were projected back onto the original UMAP to create a detailed atlas of the cell types in healthy and ruptured quadriceps tendon (Supporting information, Supplementary Fig. S10).

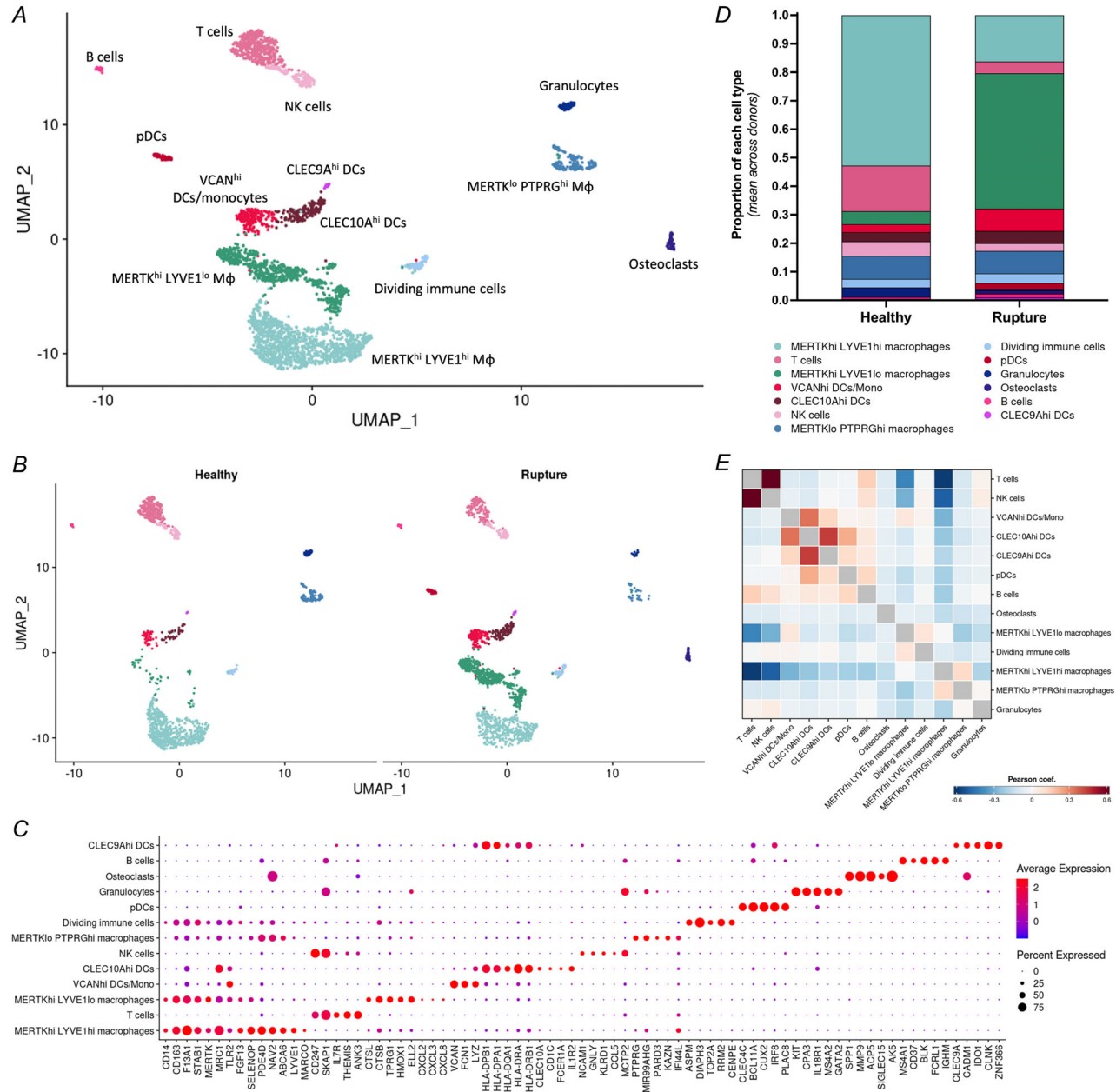

**Figure 5. Immune cell landscape in quadriceps tendon**
*A* and *B*, UMAP embedding of sub-clustering of 3178 nuclei identified as immune cells, labelled by their new cluster identity (*A*) and split by disease status (*B*). *C*, dotplot with canonical markers and differentially expressed genes used to identify the immune cell clusters. *D*, relative proportions of immune cell clusters in healthy and rupture samples. *E*, correlation matrix of immune cell clusters. DCs, dendritic cells; MΦ, macrophages. [Colour figure can be viewed at wileyonlinelibrary.com]

To understand how tendon rupture changes cell activity, we investigated potential ligand–receptor interactions between cell types in both healthy and ruptured quadriceps tendons using the CellPhoneDB database, and the activity of intracellular pathways within the healthy or diseased samples using the PROGENy database (Fig. 6*A*). CellPhoneDB is a database of ligands, receptors and their interactions. Based on the expression of these ligands and receptors across the different cell types, potential interactions can be inferred. Overall, the results showed an increase in number of potential interactions within and between the cell types in the rupture samples compared to the healthy samples. While immune cells showed some potential interactions, the cell–cell interactions occurred mainly between the different fibroblast and VEC subsets, both in the healthy and ruptured states. Pathway activity inference (Fig. 6*B*) showed an activation of TGF-$\beta$ signalling in VEC and fibroblast clusters in ruptured quadriceps tendons and an increase in VEGF signalling across immune and stromal clusters. In addition, a decrease in Janus kinase (JAK)/signal transducer and activator of transcription (STAT) signalling in ruptured quadriceps in NR4A1[hi] fibroblasts, VEC clusters, as well as several immune cell clusters was observed.

## Discussion

This study aimed to better understand the early response to human tendon injury by exploring the cellular composition and activity of acutely ruptured compared to healthy human quadriceps tendons. Fibroblasts are the predominant cell type in tendons and, although the acute injury response in tissues has often been described at the level of immune populations, much less is known about the full repertoire of cell–cell interactions that drive repair of fibroblast-rich tissues like tendon.

Clustering and sub-clustering of snRNA-seq data revealed 29 cell subsets in human quadriceps tendons, including several fibroblast, endothelial, and mural cell subsets, a plethora of immune cell subsets, as well as osteoblasts, adipocytes, and nervous system cells. Our data suggest that stromal cells – mainly fibroblast, VEC and mural cell subsets – are the main regulators of tissue homeostasis in healthy quadriceps tendons. In acutely injured quadriceps tendons, fibroblast, VEC and mural cells acquire an activated phenotype and are the main drivers of the injury response; moreover, these cells have the highest number of differentially expressed genes between healthy and rupture samples. Pathway analysis of these genes revealed EMT as a significantly increased pathway in all three cell types. Previous studies indicate that EMT is one of the most important sources of mesenchymal cells that participate in tissue repair and pathological processes (Li et al., 2016; Yang et al., 2020).

In addition, EMT has also been shown to be upregulated in human tendon-derived cells in response to stiffness, showing a potential relationship with mechanosensing (Hussien et al., 2023). Enrichment of EMT-associated genes and pathways within fibroblasts, VECs, and mural cells of ruptured quadriceps tendon suggests that these cells may have key roles in the proliferative phase of the tendon reparative response to rupture, potentially through migratory capacity, elevated resistance to apoptosis, and increased production of ECM components (Kalluri & Weinberg, 2009).

Activation of an EMT or reparative phenotype in the ruptured quadriceps tendons analysed in this study is clearly demonstrated by the shift in fibroblast states between healthy and rupture samples. In quadriceps tendon, three main fibroblast clusters were found: FBLN1[hi] fibroblasts, ABCA10[hi] fibroblasts and NR4A1[hi] fibroblasts. These three fibroblast subsets have a regulatory phenotype, with enhanced pathways including regulation of cell migration, cell adhesion, and response to chemical, respectively. NR4A1[hi] fibroblasts had high expression of nuclear receptor subgroup 4 alpha 1 gene (*NR4A1*), which has been described as a modulator of inflammation-associated fibrosis in the intestine by dampening fibrogenic signalling in myofibroblasts: activation of NR4A1 in human myofibroblasts reduced TGF-$\beta$1-induced collagen deposition and fibrosis-related gene expression (Venu et al., 2021). Therefore, the decrease of this cell type or cell state might contribute to the fibrotic response of the tissue to the injury.

Two fibroblast subsets were predominantly found in ruptured quadriceps tendons: dividing fibroblasts and ADAM12[hi] fibroblasts – which had high expression of genes, including *ADAM12*, *POSTN*, *COL1A1*, *COL1A2*, *COL3A1* and *FAP*. Several of these highly expressed genes have previously been linked to fibrosis or excessive ECM deposition, which fits with the EMT phenotype in the stromal cells of ruptured quadriceps tendon as described earlier. Several other studies have reported on ADAM12[+] fibroblasts in models of acute injury. A study by Dulauroy et al. (2012) reported that ADAM12[+] stromal cells are activated upon acute injury in the muscle and dermis. Using *in vivo* models and inducible genetic fate mapping, these injury-induced ADAM12[+] cells were shown to be specific progenitors of a major fraction of collagen-overproducing cells that are generated during scarring and that are progressively eliminated during healing (Dulauroy et al., 2012). Genetic ablation of ADAM12[+] cells was sufficient to limit generation of profibrotic cells and accumulation of interstitial collagen. Furthermore, in a study on human tendon adhesion, ADAM12[+] mesenchymal cells were found to be increased in the tendon sheath 10 days post flexor tendon laceration injury (Zhang et al., 2024). The gene expression profile of these ADAM12[+] cells showed enrichment of collagen

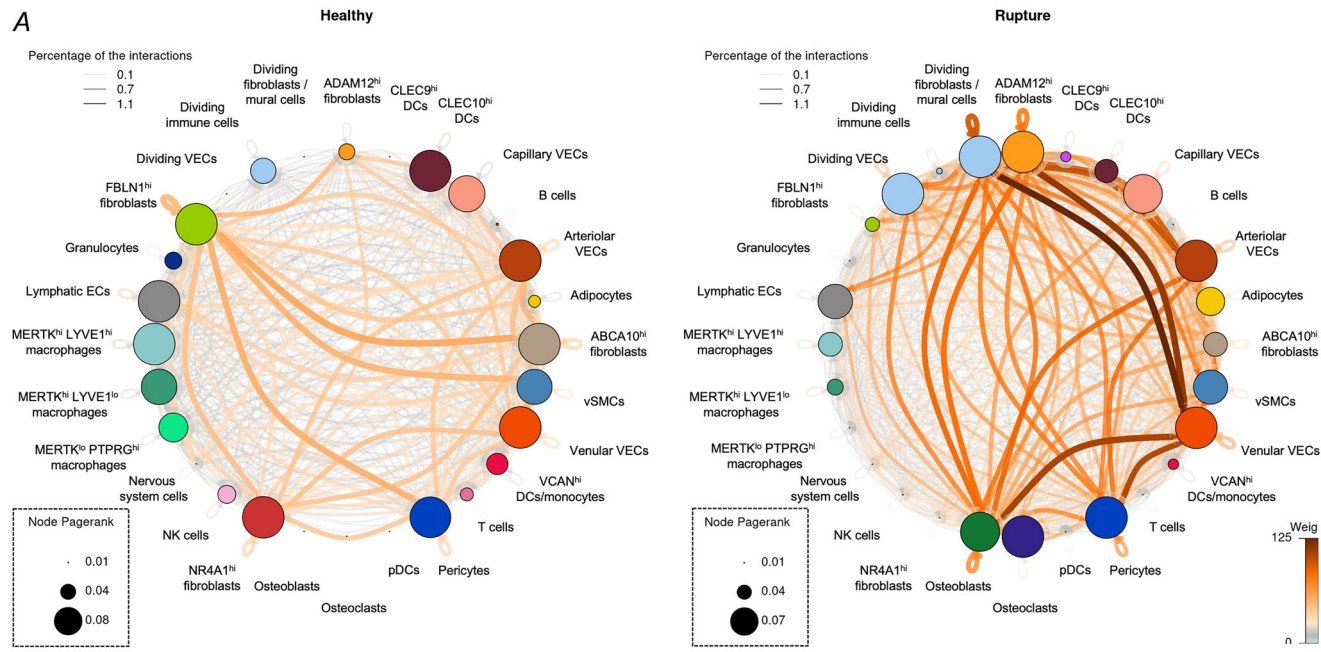

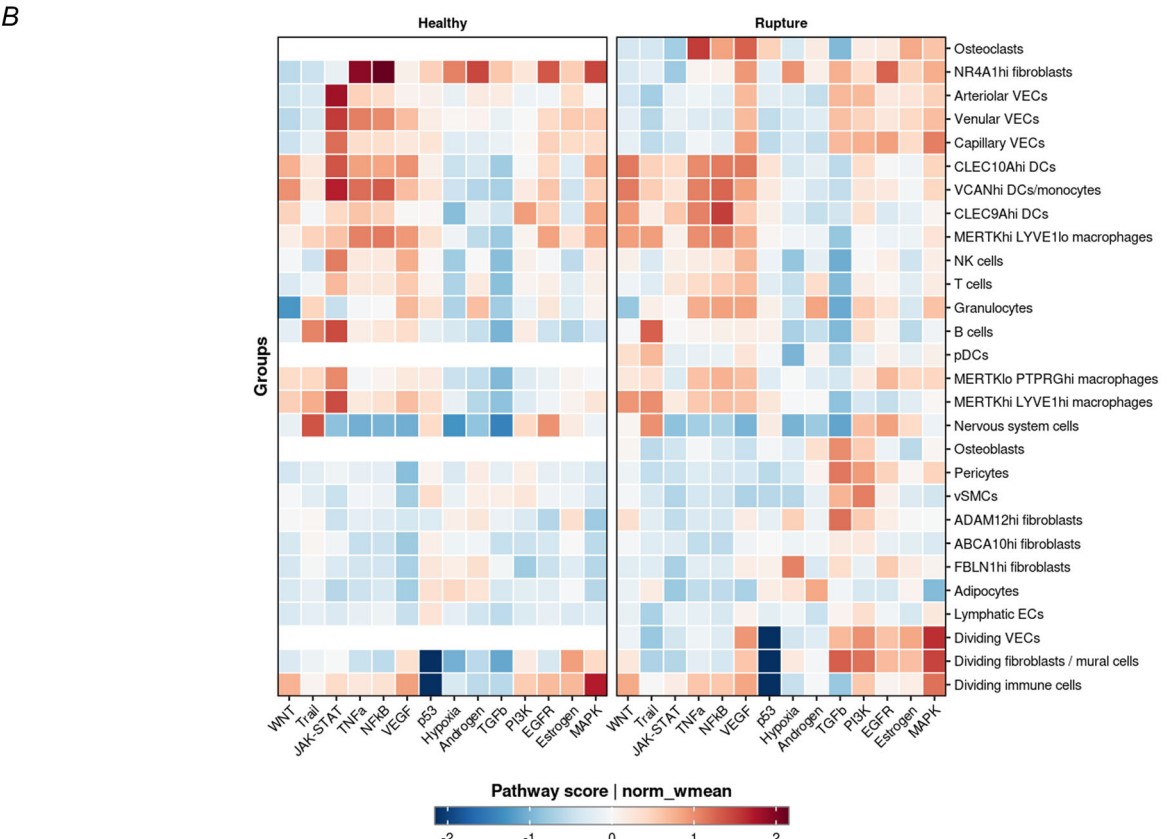

**Figure 6. Ruptured quadriceps tendons show an increase in inter-cellular associations**

*A*, ligand–receptor analysis (CellPhoneDB) results of cells in healthy (left) and ruptured (right) quadriceps tendons. Colours and thickness of connectors represent low (grey) or high (brown) number of interactions. Pagerank indicates the node importance (indicated by node size). *B*, pathway activity inference analysis results, showing the pathway activity score as the normalised weighted mean (norm_wmean). [Colour figure can be viewed at wileyonlinelibrary.com]

fibril organisation and ECM organisation, similar to the ADAM12[hi] fibroblasts in this study. This shared expression profile suggests that a common stromal-cell response may occur in the early stages post-injury.

Studies of chronic inflammatory diseases have also reported similar fibroblasts subsets to the ADAM12[hi] fibroblasts found in this study. POSTN[+] fibroblasts in rheumatoid arthritis synovium, which were reported to be high in POSTN and collagen genes, have been linked to the 'fibroid' pathotype of rheumatoid arthritis (Micheroli et al., 2022). In a skin tissue dataset, a fibroblast cluster expressing *POSTN*, *ADAM12,* and *NREP* was found to be higher in scleroderma than normal skin tissue samples (Deng et al., 2021). Using SCENIC, an analysis technique that determines which groups of transcription factors and their associated target cofactors (regulons) are significantly enriched and active within each individual cell, we showed that these fibroblasts, as well as osteoblasts and dividing fibroblasts/mural cells, had high regulon activity for TWIST1, SOX4, RARB (Supporting information, Supplementary Fig. S11), which are all involved in embryonic development, as well as in EMT and fibroblast activation, determination of cell fate and wound repair, and cell growth and differentiation, respectively (Miao et al., 2019; Qin et al., 2011; Ragavi et al., 2023; Tiwari et al., 2013; Yeo et al., 2018). Altogether, this suggests that ADAM12[hi] fibroblasts are a fibrotic cell type or state that could be a critical player in both the chronic and acute response to injury.

Angiogenesis is a crucial process in tissue repair, facilitating the delivery of oxygen and nutrients, removing waste products and controlling the immune responses (Liu et al., 2021). In ruptured quadriceps tendons, we observed a relative increase in capillary VECs, dividing VECs, and pericytes, suggesting an active angiogenesis response in the early phases following tendon injury. Analysis of pro-angiogenic factor gene expression did not reveal large differences between healthy and rupture samples, suggesting that the source of pro-angiogenic factors might be external to the tendon proper. A further pro-angiogenic cue might be via the expression of ECM proteins by rupture-regulated fibroblast subsets, with ADAM12 and POSTN known to induce pro-angiogenic signalling (Roy et al., 2017; Wasik et al., 2022), while type I collagen can support and guide endothelial cell migration (Senk & Djonov, 2021). The specific cellular events of tendon healing differ between tendons depending on the anatomy and physiology of a tendon injury and repair: healing of flexor tendons starts with angiogenesis and epitenon fibroblast migration, while in the rotator cuff fibroblasts tend to produce a disorganised collagen scar tissue and osteoclasts are attracted to the injury site (Thomopoulos et al., 2015). Therefore, the angiogenic processes in acutely ruptured quadriceps tendons require further interrogation.

This study has shown that there is a diverse immune environment in both healthy and ruptured quadriceps tendons. Tissue-resident macrophages identified in this study are similar to those identified by previous studies in tendons, as well as to those identified in other musculoskeletal tissues (Bautista et al., 2023; Sorkin et al., 2020). MERTK[hi] LYVE1[hi] macrophages have previously been shown to possess a homeostatic phenotype, with MERTK[hi] LYVE[lo] having a pro-inflammatory phenotype (Kurowska-Stolarska & Alivernini, 2022). The shift from LYVE1[hi] to LYVE1[lo] macrophages correlates with the activation of fibroblast and endothelial populations in quadriceps rupture samples in this study and has been previously observed in diseased supraspinatus tendons compared to healthy hamstring tendons (Akbar et al., 2021). This previous study also noted an increase in ECM genes, including *COL1A1* and *COL3A1*, in the stromal populations, similar to the changes described in our data earlier, suggesting some overlapping features in acute and chronic tendon tears. PTPRG[hi] macrophages have previously been hypothesised to be specialised monocyte-derived macrophages (Lissandrini et al., 2006; Mafficini et al., 2007). *MIR99AHG*, a long non-coding RNA expressed in this macrophage subset, can be increased in response to IL-4/IL-13 stimulation and has been identified as a regulator of inflammation and macrophage polarisation (Gcanga et al., 2022). Therefore, this subset might contribute to homeostasis and wound repair within quadriceps tendons. Overall, the shift of macrophages from a LYVE1[hi] to LYVE1[lo] phenotype, angiogenesis, as well as the proliferation of fibroblasts and production of collagens by fibroblasts all point towards these ruptured quadriceps tendons being towards the end of the inflammatory or beginning of the proliferative and remodelling phase of tendon healing (Leong et al., 2020; Thomopoulos et al., 2015).

Previous studies using different types of single-cell technologies on healthy and tendinopathic human tendons have identified similar broad cell types in tendons, including fibroblasts, vascular and lymphatic endothelial cells, mural cells, and some immune cells (Akbar et al., 2021; Fu et al., 2023; Karlsen et al., 2023; Kendal et al., 2020; Mimpen et al., 2024; Zhang et al., 2024). However, they have shown differences in terms of the specific subsets of cell types as well as in relative cell type abundances between healthy and tendinopathic tendons, and between tendon types. For example, compared to our previously published dataset on healthy hamstring tendon, which was produced using the same methods, healthy quadriceps tendon contained a higher abundance and variety of immune cells than hamstring (Mimpen et al., 2024), suggesting that not all tendons are the same. Future work should address this by doing a full integration of single cell transcriptomics data on all these tendon types and states.

The exploratory nature of this study means that it has a low sample number and a significant difference in age between the healthy and ruptured patient cohort. However, the similarities in gene expression profiles of the overall response and the identified cell types compared to previous studies on other injury and fibrotic pathologies support the conclusions from our data. Osteoblasts, osteoclasts, dividing VECs, and pDCs were exclusively identified in rupture samples. The presence of osteoblasts and osteoclasts in rupture samples might indicate the early stages of heterotopic ossification, which has been described to happen after injury (Zhang et al., 2020). However, the small number of cells identified, which mainly originated from one patient, means that more samples are needed to investigate this in more detail. An increase in sample number as well as a comparison with other acutely ruptured tendons, for example Achilles tendon, could further enhance our understanding of this acute injury process in human tendons. In addition, future comparisons to other tendon states, such as aged tendon, tendinopathic untorn tendons, or chronic tendon tears, could help delineate pre-existing aged-related changes *versus* the response of tendon to disease. While some of the follow-up questions would not be ethical to investigate in humans due to the poor ability of tendon to heal, the presented data could be used to validate animal models, which could investigate the long-term effects of acute ruptures and repair. Future work should also functionally characterise the fibroblasts in this study as orchestrators of the early response to tendon injury, and address whether manipulation of fibroblast populations and activities can modulate the long-term repair response. Finally, future studies would benefit from the inclusion of spatial approaches, in particular spatial transcriptomics at single cell resolution, to get more insight into the spatial niches these cells occupy and to better evaluate the potential ligand–receptor interactions that have been identified in this work.

In conclusion, this study explored the cellular landscape of healthy and ruptured quadriceps tendons. While a plethora of cell types and subsets was identified, our data indicate that fibroblasts and endothelial cells are the main drivers of the early injury response within ruptured quadriceps tendon. These cell types make up the majority of the cells within both healthy and ruptured quadriceps tendon, show the highest number of cell–cell interactions, and shift to an activated phenotype following rupture, pointing towards a fibrotic and angiogenic response. Therefore, these activated stromal cell types are obvious targets for interventions to enhance tendon healing. Overall, this study highlights multiple knowledge gaps and the need to better understand the unique biology and mechanical landscape of tendon tissue if more effective therapeutic options are to be developed.

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

## Additional information

### Data availability statement

All the code used for the data analysis of this paper is available from: https://github.com/Botnar-MSK-Atlas/quadriceps_tendon_atlas/. All the transcriptomics data in this manuscript is available to view and download from CELLxGENE: https://cellxgene.cziscience.com/collections/579203e2-182f-47bc-8230-7aa47247e2a4.

### Competing interests

A.P.C. is a cofounder and employee of Caeruleus Genomics Ltd and is an inventor on several patents related to sequencing technologies filed by Oxford University Innovations.

### Author contributions

Conceptualisation: S.J.B.S., A.P.C. and M.J.B.; methodology: J.Y.M. and C.P.; validation, C.J.C.; formal analysis: J.Y.M.; investigation: J.Y.M., M.J.B., L.R.-M., C.P., A.K., P.A.H., S.S. and M.S.N.C.F.; resources: S.E.G., M.L.C., A.M., A.T. and J.McM.; data curation: J.Y.M. and C.J.C.; writing – original draft: J.Y.M., M.J.B., S.J.B.S.; writing – review and editing, A.P.C., S.G., C.J.C., S.S., P.A.H., A.K., C.P., M.S.N. C.F.; visualisation: J.Y.M.; supervision: S.J.B.S., A.P.C. and M.J.B.; project administration: J.Y.M. and S.J.B.S.; funding acquisition: S.J.B.S., A.P.C., M.J.B., S.G. and P.A.H. All authors have read and approved the final version of this manuscript and agree to be accountable for all aspects of the work in ensuring that questions related to the accuracy or integrity of any part of the work are appropriately investigated and resolved. All persons designated as authors qualify for authorship, and all those who qualify for authorship are listed.

### Funding

This research was funded by the Chan Zuckerberg Initiative (2019-002426; J.Y.M., L.R.M., M.J.B., A.P.C., S.J.B.S.), https://chanzuckerberg.com/. J.Y.M. is funded by Versus Arthritis (22873), https://www.versusarthritis.org/. C.P. is supported by a Rhodes Scholarship (Rhodes Trust), https://www.rhodeshouse.ox.ac.uk. A.K. receives funding from the Oxford-Medical Research Council Doctoral Training Partnership, https://www.medsci.ox.ac.uk/study/graduateschool/mrcdtp/. M.L.C., S.E.G., M.J.B and S.J.B.S. were funded by and this research was supported by National Institute for Health Research (NIHR), https://www.nihr.ac.uk/ and the NIHR Oxford Biomedical Research Centre (BRC), https://oxfordhealthbrc.nihr.ac.uk/ (NIHR203311). The views expressed are those of the authors and not necessarily those of the NHS, the NIHR or the Department of Health. C.J.C. and S.S. were supported by the Chan Zuckerberg Initiative (2021-240342). A.P.C. is supported by a Medical Research Council Career Development Fellowship (MR/V010182/1), https://www.ukri.org/councils/mrc/. P.A.H. has funding from the Paget's Association (PA21010), https://paget.org.uk/. The funders had no role in study design, data collection and analysis, decision to publish, or preparation of the manuscript.

### Acknowledgements

The authors thank our research assistant Louise Appleton, our research nurses Debra Beazley and Lois Vesty-Edwards, and the surgical teams at the Nuffield Orthopaedic Centre and the John Radcliffe Hospital for their invaluable help collecting human tissue for this study. The authors would like to thank the patients for donating their tissue to research. Finally, the authors thank all other members of the CZI Tendon Seed Network for insightful discussions.

### Keywords

cellular response, endothelial cells, fibroblasts, single nucleus RNA sequencing, tendon injury, tendon repair, transcriptomics

## Supporting information

Additional supporting information can be found online in the Supporting Information section at the end of the HTML view of the article. Supporting information files available:

**Peer Review History**
**Supplementary Figure S1**
**Supplementary Figure S2**
**Supplementary Figure S3**
**Supplementary Figure S4**
**Supplementary Figure S5**
**Supplementary Figure S6**
**Supplementary Figure S7**
**Supplementary Figure S8**
**Supplementary Figure S9**
**Supplementary Figure S10**
**Supplementary Figure S11**

