## [Peer Review History · The Journal of Physiology]

Exploring cellular changes in ruptured human quadriceps tendons at single-cell resolution

Jolet Yvette Mimpfen, Mathew J Baldwin, Claudia Paul, Lorenzo Ramos-Mucci, Alina Kurjan, Carla J Cohen, Shreeya Sharma, Marie S.N. Chevalier Florquin, Philippa Hulley, John McMaster, Andrew Titchener, Alexander Martin, Matthew L Costa, Stephen E Gwilym, Adam P Cribbs, and Sarah Snelling

DOI: 10.1113/JP287812

Corresponding author(s): Jolet Mimpfen (jolet.mimpfen@ndorms.ox.ac.uk)

The following individual(s) involved in review of this submission have agreed to reveal their identity: Chloé Yeung (Referee #2)

Review Timeline:

Submission Date:	04-Oct-2024
Editorial Decision:	14-Jan-2025
Revision Received:	05-Feb-2025
Accepted:	21-Feb-2025

Senior Editor: Karyn Hamilton

Reviewing Editor: Martino Franchi

Transaction Report:

Dear Dr Mimpfen,

Re: JP-RP-2024-287812 "Exploring cellular changes in ruptured human quadriceps tendons at single-cell resolution" by Jolet Yvette Mimpfen, Mathew J Baldwin, Claudia Paul, Lorenzo Ramos-Mucci, Alina Kurjan, Carla J Cohen, Shreeya Sharma, Marie S.N. Chevalier Florquin, Philippa Hulley, John McMaster, Andrew Titchener, Alexander Martin, Matthew L Costa, Stephen E Gwilym, Adam P Cribbs, and Sarah Snelling

Thank you for submitting your manuscript to The Journal of Physiology. It has been assessed by a Reviewing Editor and by 2 expert referees and we are pleased to tell you that it is potentially acceptable for publication following satisfactory major revision.

REVISION CHECKLIST:

We look forward to receiving your revised submission.

Yours sincerely,

Karyn Hamilton
Senior Editor
The Journal of Physiology

EDITOR COMMENTS

Reviewing Editor:

Comments to the Author:

Dear Authors,

First of all I would like to apologise for the time taken with this first revision. It was very difficult to secure reviewers that had specific expertise for reviewing the present article.

Now the manuscript has been reviewed by two competent referees, and as you can see, both of them found merit in the novel data presented. However, the two referees raised important points that should be addressed. Both reviewers suggested to expand the discussion section to comment on the relevance of the tissue type selected and time points chosen to answer the biological question and to discuss the limitations.

In addition, I suggest the authors to carefully address Reviewer 2's point 4 - "Table 1 details pose ethical concerns regarding patient confidentiality...".

Senior Editor:

Comments to the Author:

Thank you for submitting your manuscript for consideration by The Journal of Physiology. I will join our Reviewing Editor in expressing apologies for the amount of time it required for us to complete the peer review process. You now have detailed feedback from two Referees with expertise in this field of study. They both express substantial enthusiasm for the manuscript and indicate that it has potential to be an impactful addition to the current body of knowledge. However, they also both raise a number of major concerns that need to be addressed before we can continue to consider the manuscript for publication. We would like to invite you to respond to each point raised by the Referees and submit a manuscript revised based on their feedback. We look forward to seeing your revised work and appreciate your interest in The Journal.

REFEREE COMMENTS

Referee #1:

Mimpen, Baldwin, and colleagues explore early cellular changes within the quadriceps tendon following rupture at single cell (nucleus) resolution. Given the relatively poor healing of tendons following rupture, there is a need to identify novel early cellular changes that may reveal therapeutic points of intervention. The current study provides compelling evidence for alterations across a number of cell types within the tendon, including fibroblasts and capillaries. Overall, the manuscript is well written and the bioinformatic analysis of the snRNA-seq data are robust. I have a few relatively minor comments:

There are a number of recent single cell analyses of human tendon that the authors cite, but there is little discussion of the relatively similarities and differences between the current data and prior single cell analyses of human tendon, both in healthy and during tendon pathology.

I fully admit to not being a tendon biologist, but the one cell type I was expecting to see in the analysis was tenocytes. Can the authors comment on tenocyte absence?

Given the relatively low resolution used to perform subcluster analysis, it would be helpful to present some of the candidate biomarkers as feature plots to visually portray specificity.

The raw and tabulated data from the snRNA analysis should be deposited in a publicly-available repository, such as GEO.

It is assumed that all quadriceps tendon ruptures were complete - is this correct?

Given the weight of the predicted fibroblast and endothelial cell interactions, I was curious as to whether these 2 cell types tend to occupy the same relative niche in vivo.

Referee #2:

In this manuscript, the authors, Mimpfen, Baldwin et al. investigated the early cellular changes with ruptures, in order to generate new therapeutic targets for interventions that would enhance tendon healing. They collected human quadriceps tendons from healthy and acute rupture (7-8 days post-injury) and subjected them to single nucleus RNA sequencing analysis, based on a previously published protocol. The authors characterized the cell type profile of healthy and injured tendons, the changes in gene expression of these cell types and the pathways and the biological processes that are potentially differentially regulated with the early stages of tissue injury. They found that most of the changes occur in the fibroblast, vascular endothelial cell and mural cell populations. The manuscript reports novel findings in the understanding of cell populations in healthy and ruptured human quadriceps tendons and identifies potential cell types that could be targeted to improve tendon healing.

I have a few comments and suggestions improve the clarity and completeness of the manuscript, in particular, a justification of the chosen tendon type and injury time point, and the limitations of these choices should be discussed. Please see below:

Introduction

1. "Acute tendon ruptures often occur on the background of pre-existing tendinopathy and usually require surgical intervention or prolonged immobilisation." This line in the second introduction paragraph requires a reference, as it is still a matter of debate whether tendinopathy leads to ruptures.

2. "Similarly, the role of the immune, endothelial and mural cell compartments has been investigated in the context of chronic tendinopathy in humans and rarely during the early injury response[10]" - Line on bottom of page 3 should specify that the injury referred to is an acute rupture (and not early stages of tendinopathy). Additionally, the reference 10 ("Light microscopic histology of quadriceps tendon ruptures") does not seem to be relevant for this statement.

3. In the Key Points and Abstracts, the authors state that ruptures have been regularly studied during the chronic phase of

injury. Can the authors expand on this in the introduction and state the limitations/knowledge gaps of these studies? Highlighting these points would allow the reader to appreciate the new knowledge generated by the current study.

Methods section

4. Table 1 details pose ethical concerns regarding patient confidentiality. While it does not include explicit identifiers like names or dates of birth, the combination of information (e.g., age, sex, ethnicity, diagnosis, and surgery type) can make patients potentially identifiable (especially for themselves, particularly in smaller populations or specific medical contexts). If inclusion of the table is essential, then the data needs to have adequate anonymisation, e.g., removal of IDs (also from Table 2) and use of broader age ranges or report the mean and SD of the two groups.

5. Table 2, perhaps just report the samples that were analysed.

Results

6. From the methods section, the reader understood that the samples were: 5 healthy (4 analysed), 3 rupture (3 analysed). To avoid confusion, the first line in results should be reworded to say that 4 healthy and 3 ruptures were analysed rather than collected, as one more sample was initially collected for healthy group.

7. Although it is mentioned later in the discussion, the significant difference in age between the healthy and ruptured samples should be reported here in the results section for transparency.

8. Due to the uneven distribution of nuclei from each sample analysed, can the authors comment or perhaps include data to show the distribution of cell types present in the replicate samples within each group? It is unclear whether a certain cell type is overrepresented in one sample compared to another and indeed, the authors mention later in the discussion that the presence of bone cells are mostly contributed by one sample.

Discussion

9. Quadriceps ruptures are rare and often occur with comorbidities. Can the authors comment on why this tendon was chosen for the study? And comment in the Methods section, whether there were any exclusion criteria for the donors of the tissues.

10. The rationale behind the experimental design and its potential limitations could be more thoroughly addressed. Specifically, how does investigating the early rupture response in quadriceps tendons translate into insights for improved healing and preventing re-ruptures that occur more frequently in other tendon types, such as the Achilles tendon? Would an investigation into how certain early injury cellular profiles predispose tendons to reinjury, or an analysis of tissues that have already re-injured, offer a more direct approach?

Expanding on these points in the discussion would strengthen the study's broader relevance.

11. In line with relating data reported in one tendon type for potential therapeutic implications in another, can the authors briefly comment on the overall similarities or differences observed in their current snRNAseq data with their previous work in hamstring tendons (Mimpen JY, Ramos-Mucci L, Paul C, Kurjan A, Hulley PA, Ikwuanusi1 CT, Cohen CJ, Gwilym SE, Baldwin MJ, Cribbs AP, Snelling SJB (2024) Single nucleus and spatial transcriptomic profiling of healthy human hamstring tendon. *The FASEB Journal* 38:e23629).

Figures

12. Reference to Fig. 1B in text should be 1C, 1C in text should be 1D.

13. Perhaps the red/green colouring, especially in Fig. 2 and 3A, can be changed to colour-blind friendly colours.

14. Figure 2C and 3A key, what does blue for p-value refer to? Should be corrected to say $p > X$ or not highlighted at all. Same goes for green highlighted genes.

15. Figure 2D, what does the key for size refer to?

END OF COMMENTS

Oxford, 05 February 2025

Dear Reviewers and Editors,

We thank Reviewers 1 and 2 for their insightful comments and suggestions. We have provided a response to each comment below. All changes to the manuscript are stated, including textual additions, which have been underlined.

Best wishes,
Dr Jolet Mimpen
on behalf of all authors

REFeree COMMENTS

Referee #1:

Mimpen, Baldwin, and colleagues explore early cellular changes within the quadriceps tendon following rupture at single cell (nucleus) resolution. Given the relatively poor healing of tendons following rupture, there is a need to identify novel early cellular changes that may reveal therapeutic points of intervention. The current study provides compelling evidence for alterations across a number of cell types within the tendon, including fibroblasts and capillaries. Overall, the manuscript is well written and the bioinformatic analysis of the snRNA-seq data are robust. I have a few relatively minor comments:

There are a number of recent single cell analyses of human tendon that the authors cite, but there is little discussion of the relatively similarities and differences between the current data and prior single cell analyses of human tendon, both in healthy and during tendon pathology.

Authors' response: we thank reviewer 1 for their comment. We have now added the following paragraph to the Discussion section: *“Previous studies using different types of single-cell technologies on healthy and tendinopathic human tendons have identified similar broad cell types in tendons, including fibroblasts, vascular and lymphatic endothelial cells, mural cells, and some immune cells[5–9, 41]. However, they have shown differences in terms of the specific subsets of cell types as well as in relative cell type abundances between healthy and tendinopathic tendons, and between tendon types. For example, compared to our previously published dataset on healthy hamstring tendon, which was produced using the same methods, healthy quadriceps tendon contained a higher abundance and variety of immune cells than hamstring[5], suggesting that not all tendons are the same. Future work*

should address this by doing a full integration of single cell transcriptomics data on all these tendon types and states.”

I fully admit to not being a tendon biologist, but the one cell type I was expecting to see in the analysis was tenocytes. Can the authors comment on tenocyte absence?

Authors’ response: tenocytes are a term sometimes used for the resident cells in tendons, namely tendon fibroblasts. We chose not to use the term tenocyte in this manuscript as tenocyte makes readers assume that these cells (and cell subsets) are specific to tendons, while single cell-resolution transcriptomics studies have generally shown many similarities in fibroblasts across different tissues and organ systems. However, to clarify the term, we have added the following clarification to the introduction: “Fibroblasts, in tendon sometimes referred to as tenocytes, are responsible for ECM production and maintenance in tendon development, homeostasis and repair[3, 7].”

Given the relatively low resolution used to perform subcluster analysis, it would be helpful to present some of the candidate biomarkers as feature plots to visually portray specificity.

Authors’ response: we assume that Reviewer would like us to add feature plots of differentially expressed genes of each fibroblast subcluster. We have now added these as Supplementary Figure 7, in which three genes per subclusters are shown. The other Supplementary Figures have been renumbered accordingly.

The raw and tabulated data from the snRNA analysis should be deposited in a publicly-available repository, such as GEO.

Authors’ response: we thank reviewer 1 for their comment. We apologise if this was not made clear, but all the data has been uploaded to CELLxGENE and the links will be made publicly available as soon as the paper is accepted for publication.

It is assumed that all quadriceps tendon ruptures were complete - is this correct?

Authors’ response: this is correct, surgery is only indicated in our institution for full ruptures. We have added the following statement to our methods section for clarification: “Ruptured quadriceps tendon tissue was collected from patients undergoing tendon repair surgery for complete quadriceps tendon rupture 7-8 days post injury.”

Given the weight of the predicted fibroblast and endothelial cell interactions, I was curious as to whether these 2 cell types tend to occupy the same relative niche in vivo.

Authors’ response: we thank reviewer 1 for their question and agree that this is an interesting point. We agree that these predicted interactions are better understood in combination with spatial data. This is why we have used the word “potential” ligand-receptor interaction. Unfortunately, the small samples obtained for this study have been completely exhausted, meaning that we will not be able to add this to the current manuscript. However, we hope to pursue this in more detail in the future. Therefore, the following sentence has been added to the limitations/future work section in the Discussion: “Finally, future studies would benefit from the inclusion of spatial approaches, in

particular spatial transcriptomics at single cell resolution, to get more insight into the spatial niches these cells occupy and to better evaluate the potential ligand-receptor interactions that have been identified in this work.”.

Referee #2:

In this manuscript, the authors, Mimpfen, Baldwin et al. investigated the early cellular changes with ruptures, in order to generate new therapeutic targets for interventions that would enhance tendon healing. They collected human quadriceps tendons from healthy and acute rupture (7-8 days post-injury) and subjected them to single nucleus RNA sequencing analysis, based on a previously published protocol. The authors characterized the cell type profile of healthy and injured tendons, the changes in gene expression of these cell types and the pathways and the biological processes that are potentially differentially regulated with the early stages of tissue injury. They found that most of the changes occur in the fibroblast, vascular endothelial cell and mural cell populations. The manuscript reports novel findings in the understanding of cell populations in healthy and ruptured human quadriceps tendons and identifies potential cell types that could be targeted to improve tendon healing.

I have a few comments and suggestions improve the clarity and completeness of the manuscript, in particular, a justification of the chosen tendon type and injury time point, and the limitations of these choices should be discussed. Please see below:

Introduction

1. "Acute tendon ruptures often occur on the background of pre-existing tendinopathy and usually require surgical intervention or prolonged immobilisation." This line in the second introduction paragraph requires a reference, as it is still a matter of debate whether tendinopathy leads to ruptures.

Authors' response: we thank reviewer 2 for their valid comment. Quadriceps tendon ruptures do not happen in younger patients, regardless of the injury type. The same type of injury will lead to patellar tendon rupture or patellar fracture before quads rupture in younger patients. However, we agree that this is still a hypothesis which has not been fully proven. Therefore, we have added two references and amended the said sentence to the following: "Acute tendon ruptures are hypothesised to occur on the background of pre-existing tendinopathy and usually require surgical intervention or prolonged immobilisation[2, 3]."

2. "Similarly, the role of the immune, endothelial and mural cell compartments has been investigated in the context of chronic tendinopathy in humans and rarely during the early injury response[10]" - Line on bottom of page 3 should specify that the injury referred to is an acute rupture (and not early

stages of tendinopathy). Additionally, the reference 10 ("Light microscopic histology of quadriceps tendon ruptures") does not seem to be relevant for this statement.

Authors' response: we have amended the sentence at the bottom of page 3 to specify that this is about an acute tendon injury: "Similarly, the role of the immune, endothelial and mural cell compartments has been investigated in the context of chronic tendinopathy in humans and rarely during the early injury response of an acute tendon injury[2, 7, 8].".
Regarding the cited paper: this paper has looked at the vascularity of ruptured quadriceps tendons and has compared this to control quadriceps tendons from cadavers, which we deem relevant due to the mention of the endothelial compartment in this sentence. However, we have added two more references of single cell studies that investigated tendinopathy in supraspinatus tendon.

3. In the Key Points and Abstracts, the authors state that ruptures have been regularly studied during the chronic phase of injury. Can the authors expand on this in the introduction and state the limitations/knowledge gaps of these studies? Highlighting these points would allow the reader to appreciate the new knowledge generated by the current study.

Authors' response: we thank reviewer 2 for their comment. Most tendon tears that have been studied previously are rotator cuff tears; these tears are often repaired months or even years after a non-specific injury event (patients often do not know when the injury occurred). We have made several changes to the manuscript to make it clearer what we mean with early and chronic phase of the injury, and to highlight that chronic tendinopathy and tendon tears have mostly been studied in the rotator cuff tendons of the shoulder:

- **Key points:**
 - *"Tendon ruptures in humans have regularly been studied during the chronic phase of injury, but less is known about the early injury response after acute tendon ruptures."*
- **Abstract:**
 - *"Tendon ruptures in humans have regularly been studied during the chronic phase of injury, in particular rotator cuff disease. However, the early response to acute tendon ruptures remains less investigated."*
- **Introduction:**
 - *"Fibroblast activation has been studied in human chronic tendinopathy, particularly in rotator cuff tendons of the shoulder[1, 13–15], in which tears are often atraumatic and might be surgically repaired months or years after a non-injury event[16–18]. However, the fibroblast cell types and states important in the early human response to acute tendon rupture have not been characterised well."*

Methods section

4. Table 1 details pose ethical concerns regarding patient confidentiality. While it does not include explicit identifiers like names or dates of birth, the combination of information (e.g., age, sex, ethnicity, diagnosis, and surgery type) can make patients potentially identifiable (especially for themselves, particularly in smaller populations or specific medical contexts. If inclusion of the table is essential, then the data needs to have adequate anonymisation, e.g., removal of IDs (also from Table 2) and use of broader age ranges or report the mean and SD of the two groups.

Authors' response: we appreciate reviewer's 2 concern regarding patient confidentiality. This issue has been raised several times and discussed in published works, including in Knoppers et al. 2023, Annual Review of Genomics and Human Genetics), which discusses the risk of deidentification methods and the risk of reidentification in detail. We have carefully considered this and spoken to our in-house experts to ensure that this data can be reported. Below, we have set out several considerations and reasons for not adjusting this table.

1. As researchers we have access to minimal metadata of each patient to ensure that the risk of de-anonymising is minimised, and all these fields have been approved.
2. The time window during which the samples were collected has not been published and neither has the hospital site, minimising the risk of de-anonymisation. In addition, thousands of patients consent to their waste tissue being used for research every year across our institution.
3. The study number is unique to our records and has no relation with hospital numbers or patient care. In addition, the patient does not know the study number that has been assigned to them and the researchers do not know the patient's identity or any identifiable information about the patient. We have clarified this by adding the following to the manuscript: *"The age, sex and self-reported ethnicity of donors, alongside the affected side of the injury, were collected and are reported along the de-identified study ID of each patient (Table 1)."* In addition, legend for Table 1 also specified this is a de-identified patient ID.
4. Finally, the metadata fields that have been shared are all included in the Human Cell Atlas list of tier 1 metadata. These fields are highly recommended to be reported in all single cell studies. Therefore, these metadata fields are generally not considered to be so specific that they cannot be made available.

We can confirm that the same patient demographics, but with the addition of BMI, has been shared in our paper on healthy hamstring paper (Mimpen *et al.* 2024, FASEB).

5. Table 2, perhaps just report the samples that were analysed.

Authors' response: we confirm that we have removed the row including the analysis parameters for MSK 1250 to avoid any confusion. The design for the two samples sequenced for patient MSK 0779 has been simplified to emphasise that these samples indeed came from one donor as thoroughly described in the Table legend.

Results

6. From the methods section, the reader understood that the samples were: 5 healthy (4 analysed), 3 rupture (3 analysed). To avoid confusion, the first line in results should be reworded to say that 4 healthy and 3 ruptures were analysed rather than collected, as one more sample was initially collected for healthy group.

Authors' response: as detailed in Table 1: 4 healthy and 3 rupture samples were collected; however, data from one of the healthy samples could not be included due to high contamination of skeletal muscle cells. MSK 1250 has now been removed from Table 2 to avoid any confusion about its inclusion in the final dataset. The two samples for MSK 0779

(one patient for which 2 samples were run to due low yield as explained in Table legend) have now been merged to make it clearer that they are from the same patient.

7. Although it is mentioned later in the discussion, the significant difference in age between the healthy and ruptured samples should be reported here in the results section for transparency.

Authors' response: the basic patient details (gender and age range) have now been added to the Results section: *"After extensive quality control, the transcriptomes of 12,808 nuclei were profiled, including 7,268 nuclei from three healthy donors (male, 25-44 years old) and 5,540 nuclei from three ruptured quadriceps tendon donors (male, 67-75 years old)."*

8. Due to the uneven distribution of nuclei from each sample analysed, can the authors comment or perhaps include data to show the distribution of cell types present in the replicate samples within each group? It is unclear whether a certain cell type is overrepresented in one sample compared to another and indeed, the authors mention later in the discussion that the presence of bone cells are mostly contributed by one sample.

Authors' response: The distribution of cell types present in each of the samples can be found in Supplementary Figure 5, in which UMAP and bar charts for each individual sample are displayed. The overall trends are similar, which is mentioned in the first sentence of the second paragraph of the Results section *"Differences were observed in the mean cell type frequency between healthy and ruptured quadriceps tendons (Figure 1B), and the trends in cell type abundance were similar across patients (Supplementary Figure 5)."*

Discussion

9. Quadriceps ruptures are rare and often occur with comorbidities. Can the authors comment on why this tendon was chosen for the study? And comment in the Methods section, whether there were any exclusion criteria for the donors of the tissues.

Authors' response: although the incidence of quadriceps ruptures is rare compared to other diseases, quadriceps tendons are the most common tendon to undergo repair for acute rupture. Therefore, this tendon was chosen for this work. We have also added this to the last paragraph of the Introduction: *"Quadriceps tendons were used for this study, since they are one of the most common tendons to undergo repair for acute rupture."*

Regarding the exclusion criteria. The only exclusion criterium in this work was "active infection at time of surgery". This has now been added to the Methods section: *"Patients were not included if they had an active infection at time of surgery."*

10. The rationale behind the experimental design and its potential limitations could be more thoroughly addressed. Specifically, how does investigating the early rupture response in quadriceps tendons translate into insights for improved healing and preventing re-ruptures that occur more frequently in other tendon types, such as the Achilles tendon? Would an investigation into how certain early injury cellular profiles predispose tendons to reinjury, or an analysis of tissues that have

already re-injured, offer a more direct approach? Expanding on these points in the discussion would strengthen the study's broader relevance.

Authors' response: we thank reviewer 2 for their suggestions and appreciate the need for additional discussion on the study's relevance. We have been careful with our statements on this so far, since we do not want to over-conclude on the presented data or its translatability. However, this dataset does offer a new perspective into the continuum from acute rupture to repair. While it would not be ethical to investigate most follow-up questions in humans, these findings could be used to validate animal models, which – in turn – could be used to investigate follow-up questions on the long-term effects of acute ruptures and repair. We have added several statements about this in the Discussion:

“While some of the follow-up questions would not be ethical to investigate in humans due to the poor ability of tendon to heal, the presented data could be used to validate animal models, which could investigate the long-term effects of acute ruptures and repair”.

Regarding the question about Achilles tendon: Achilles tendon would represent another potential tendon injury model. However, Achilles tendon ruptures are now rarely surgically repaired in the United Kingdom, so there is no human tissue available for us to include in research.

11. In line with relating data reported in one tendon type for potential therapeutic implications in another, can the authors briefly comment on the overall similarities or differences observed in their current snRNAseq data with their previous work in hamstring tendons (Mimpen JY, Ramos-Mucci L, Paul C, Kurjan A, Hulley PA, Ikwuanusi CT, Cohen CJ, Gwilym SE, Baldwin MJ, Cribbs AP, Snelling SJB (2024) Single nucleus and spatial transcriptomic profiling of healthy human hamstring tendon. The FASEB Journal 38:e23629).

Authors' response: thank you for your question. We have added a paragraph into the Discussion that highlights some similarities and differences between this study and previously published studies using single cell technologies on human tendons, including our own study on healthy hamstring tendons. : *“Previous studies using different types of single-cell technologies on healthy and tendinopathic human tendons have identified similar broad cell types in tendons, including fibroblasts, vascular and lymphatic endothelial cells, mural cells, and some immune cells[5–9, 41]. However, they have shown differences in terms of the specific subsets of cell types as well as in relative cell type abundances between healthy and tendinopathic tendons, and between tendon types. For example, compared to our previously published dataset on healthy hamstring tendon, which was produced using the same methods, healthy quadriceps tendon contained a higher abundance and variety of immune cells than hamstring[5], suggesting that not all tendons are the same. Future work should address this by doing a full integration of single cell transcriptomics data on all these tendon types and states.”*

We believe that a full integration of transcriptomics data of all tendon types is now required to answer the question raised by reviewer 2; however, this is the beyond the scope of this work.

Figures

12. Reference to Fig. 1B in text should be 1C, 1C in text should be 1D.

Authors' response: thank you for pointing these out. The references to Figure 1 have now been amended.

13. Perhaps the red/green colouring, especially in Fig. 2 and 3A, can be changed to colour-blind friendly colours.

Authors' response: all the red/green colouring that refers to healthy (green) or rupture (red) samples has now been changed to a colourblind friendly colour combination of purple (healthy) and orange (rupture). These changes were made in Figures 1-4. The colours of the volcano plots were not changed due to this being a standard colour palette for these graphs and there being guides (dotted lines) to point out the log₂ fold change and p-value thresholds.

14. Figure 2C and 3A key, what does blue for p-value refer to? Should be corrected to say p>X or not highlighted at all. Same goes for green highlighted genes.

Authors' response: this is a standardised way of displaying results for volcano plots in which blue refers to genes that matched the p-value threshold but not the log₂ fold change threshold, and green refers to genes that matched the log₂ fold change threshold but not the p-value threshold. We have amended the Figure 2C legends to explain this: "Colours: not-significant genes (NS; grey), genes that reached the log₂ fold change threshold, but not the p-value threshold (green), genes that reached the p-value threshold but not the log₂ fold change threshold (blue), or genes that were considered significant due to reaching both the p-value and log₂ fold change threshold (red).". Legend for Figure 3A-B was also amended accordingly: "(A) Colours refer to not-significant genes (NS; grey), genes that reached the log₂ fold change threshold, but not the p-value threshold (green), genes that reached the p-value threshold but not the log₂ fold change threshold (blue), or genes that were considered significant due to reaching both the p-value and log₂ fold change threshold (red). (B) Colours refer to the type of matrisome component."

15. Figure 2D, what does the key for size refer to?

Authors' response: the size refers to the number of genes in each node. We have added this information to the Figure legend to clarify this: "Node size (number of genes in a category); colour (fold change of each displayed gene)."

Authors:

A few other additions were made to the manuscript:

- **Title page:**
 - **A reference to our pre-print has been added as per journal policy.**

- **Abstract:**
 - Several small adaptations were made to the abstract to ensure the abstract was below the 250 words limit after the additions made in response to reviewers.
- **Figure legends:**
 - Several Figure legends were changed to correctly reflect the colours representing healthy (purple) and rupture (orange) quadriceps tendon samples.
 - Figure legend 2 now states the correct number of nuclei in this dataset (12,808).
- **Discussion:**
 - The following sentence in the first paragraph of the Discussion was re-written to increase clarity. **Old sentence:** *"Fibroblasts are the predominant cell type in tendons and, although the acute injury response in tissues has often been described at the level of immune populations, much less is known about the full repertoire of cell-cell interactions that drive repair of fibroblast-rich tissues like tendon."* **New sentence:** *"Fibroblasts are the predominant cell type in tendons and, although the acute injury response in tissues has often been described at the level of immune populations, much less is known about the full repertoire of cell-cell interactions that drive repair of fibroblast-rich tissues like tendon."*
- **Supplementary Figures:**
 - Supplementary Figure 7-10 were re-numbered 8-11 to account for the addition of Supplementary Figure 7.

Dear Dr Mimpfen,

Re: JP-RP-2025-287812R1 "Exploring cellular changes in ruptured human quadriceps tendons at single-cell resolution" by Jolet Yvette Mimpfen, Mathew J Baldwin, Claudia Paul, Lorenzo Ramos-Mucci, Alina Kurjan, Carla J Cohen, Shreeya Sharma, Marie S.N. Chevalier Florquin, Philippa Hulley, John McMaster, Andrew Titchener, Alexander Martin, Matthew L Costa, Stephen E Gwilym, Adam P Cribbs, and Sarah Snelling

We are pleased to tell you that your paper has been accepted for publication in The Journal of Physiology.

- The reference list must be in alphabetical order, rather than numbered, to comply with our Journal format.

- The Journal of Physiology funds authors of provisionally accepted papers to use the premium BioRender site to create high resolution schematic figures. Follow this link and enter your details and the manuscript number to create and download figures. Upload these as the figure files for your revised submission. If you choose not to take up this offer, we require figures to be of similar quality and resolution. If you are opting out of this service to authors, state this in the Comments section on the Detailed Information page of the submission form. The link provided should only be used for the purposes of this submission. Authors will be charged for figures created on this premium BioRender account if they are not related to this manuscript submission.

Yours sincerely,

Karyn Hamilton
Senior Editor
The Journal of Physiology

If you would like to receive our 'Research Roundup', a monthly newsletter highlighting the cutting-edge research published in The Physiological Society's family of journals (The Journal of Physiology, Experimental Physiology, Physiological Reports, The Journal of Nutritional Physiology and The Journal of Precision Medicine: Health and Disease), please click this link, fill in your name and email address and select 'Research Roundup':

<https://www.physoc.org/journals-and-media/membernews>

- You can help your research get the attention it deserves! Check out Wiley's free Promotion Guide for best-practice recommendations for promoting your work at: www.wileyauthors.com/eoo/guide. You can learn more about Wiley Editing Services which offers professional video, design, and writing services to create shareable video abstracts, infographics, conference posters, lay summaries, and research news stories for your research at: www.wileyauthors.com/eoo/promotion.

EDITOR COMMENTS

Reviewing Editor:

Comments to the Author:

All comments were thoroughly taken in consideration and addressed.

I congratulate the authors for the acceptance and for a very interesting paper! Bravi!

Senior Editor:

Comments to the Author:

Thank you for your careful revisions. We are pleased to accept your manuscript for publication in The Journal of Physiology. Congratulations and thank you for your interest in The Journal!

REFEREE COMMENTS

Referee #1:

The authors have satisfactorily addressed my previous comments. I have no further comments, congrats on a great study!

Referee #2:

Thank you to the authors for their prompt revision. The authors have responded satisfactorily to all my comments. I want to congratulate them for generating this important set of data, which is crucial for advancing our understanding of tendon biology.